# Long-range terrestrial laser scanning measurements of annual and intra-annual mass balances for Urumqi Glacier No.1, eastern Tien Shan, China

Chunhai Xu[1, 2], Zhongqin Li[1], Huilin Li[1, 2], Feiteng Wang[1], and Ping Zhou[1]

[1]State Key Laboratory of Cryospheric Science/Tien Shan Glaciological Station, Northwest Institute of Eco-Environment and Resources, Chinese Academy of Sciences, Lanzhou 730000, China

[2]University of Chinese Academy of Sciences, Beijing 100049, China

**Correspondence:** Zhongqin Li (lizq@lzb.ac.cn)

**Abstract.** The direct glaciological method provides in situ observations of annual or seasonal surface mass balance, but can only
be implemented through a succession of intensive in situ measurements of field networks of stakes and snow pits. This has contributed to glacier surface mass-balance measurements being sparse and often discontinuous in the Tien Shan. Nevertheless, long-term glacier mass-balance measurements are the basis for understanding climate–glacier interactions and projecting future water availability for glacierized catchments in the Tien Shan. Riegl VZ®-6000 long-range terrestrial laser scanner (TLS), typically using class 3B laser beams, is exceptionally well suited for repeated glacier mapping, and thus determination of annual and
seasonal geodetic mass balance. This paper introduces the applied TLS for monitoring summer and annual surface elevation and geodetic mass changes of Urumqi Glacier No.1 as well as delineating accurate glacier boundaries for two consecutive mass balance years (2015-17), and discusses the potential of such technology in glaciological applications. Three-dimensional changes of ice and firn/snow bodies and the corresponding densities were considered for the volume-to-mass conversion. The glacier showed pronounced thinning and mass loss for the four investigated periods; glacier-wide geodetic mass balance in the mass-balance year
2015-16 was slightly more negative than in 2016-17. Statistical comparison shows that agreement between the glaciological and geodetic mass balances can be considered as satisfactory; indicating that the TLS system yields accurate results and has the potential to monitor remote and inaccessible glacier areas where no glaciological measurements are available as the vertical velocity component of the glacier is negligible. For wide applications of the TLS in glaciology, we should use stable scan positions and in situ measured densities of snow/firn to establish volume-to-mass conversion.

## 1 Introduction

Glacier meltwater is a crucial freshwater resource for populations and hydro-economies in arid and semi-arid regions (e.g. Sorg et al., 2012; Chen et al., 2016). The concept of 'solid reservoirs' is well represented in the Tien Shan, where most glaciers have experienced substantial mass loss over recent decades (Farinotti et al., 2015; Pieczonka et al., 2015; Liu and Liu., 2016; Sakai et al., 2017; Li et al., 2018). Hence, a better understanding of the relationship between Tien Shan glacier wastage and changing climate is
important for projecting water availability in the near future. Glacier mass balance provides important information on the gain or loss in glacier mass and is a direct and immediate indicator of climate evolution (Kaser et al., 2006; Haeberli et al., 2007).

Continuous mass-balance observations are fundamental to understand climate–glacier interactions (Zemp et al., 2015). Annual and sometimes seasonal surface mass balance of individual glaciers can be measured using the direct glaciological method. Stakes are
drilled into the ice, allowing the monitoring of ablation, and snow pits are dug in the area where snow has accumulated to provide

net accumulation (Østrem and Brugman, 1991; Xie and Liu, 1991; Cogley et al., 2011). However, the shortage of long-term financial and human resources and the inaccessibility of remote regions and natural hazards means that ongoing in situ glacier mass-balance measurements are sparse and discontinuous in the Tien Shan, only Tuyuksu glacier (northern Tien Shan, Kazakhatan) and Urumqi Glacier No.1 (eastern Tien Shan, China) have long glaciological mass balance series (Hoelzle et al., 2017). In contrast to the extensive in situ measurement networks required for glaciological observations, the geodetic method provides mass balance by repeated surveys of the glacier surface terrain, in which two digital elevation models (DEMs) are subtracted to calculate the volume changes and then convert them to mass balance using a density conversion (Zemp et al., 2013; Huss, 2013; Andreassen et al., 2016). The method includes all processes that affect the surface, internal and basal mass balances (Cuffey and Paterson, 2010; Sold et al., 2016), but the geodetic mass balances are assumed to be accurate since the topographic surveys are of high quality (Thibert et al., 2008; Huss et al., 2009; Joerg et al., 2012). The available DEMs, derived from aerial photography and traditional remote sensing imagery, usually limit the accuracy and time resolution of geodetic mass-balance measurements (e.g. Cox and March et al., 2004; Cogley, 2009; Fischer, 2011).

In recent years, emerging earth observation technologies (e.g. airborne (ALS) and terrestrial laser scanning (TLS)) allow the derivation of high-resolution DEMs with vertical and horizontal errors on the order of a few centimeters, and increasingly been used to calculate geodetic mass balance and changes in glacier volume (e.g. Geist et al., 2005; Pellikka and Rees, 2009; Joerg et al., 2012; Gabbud et al., 2015; Fischer et al., 2016; Klug et al., 2018). ALS is effective for rapidly mapping extensive areas, but the difficulty of studying glacial changing processes with high temporal resolution since the high costs of ALS and the presence of great topographic relief and high-altitude rock outcrops around glaciers reduce the capacity of observations by aircraft as most ALS instruments have limited operating flight altitude, so we need ground-based surveys (Piermattei et al., 2015). The TLS system is usually simpler, more economical and more flexible than ALS, and has become a well-established tool for monitoring annual and sometimes seasonal changes in individual glaciers (e.g. Gabbud et al., 2015; Fischer et al., 2016; López-Moreno et al., 2016). The new high-speed and high-resolution Riegl VZ®-6000 terrestrial laser scanner offers a long measurement range of more than 6 km and a wide field of 60° vertical and 360° horizontal for topographic (static) applications (RIEGL Laser Measurement Systems, 2014a). The scanner is a Laser Class 3B, with laser wavelength in the near-infrared (~1064 nm), and thus well-suited for measuring snow- and ice-covered terrain in repeated glacier mapping. One study has covered the novel use of Riegl VZ®-6000 TLS to measure surface melt for a temperate Alpine valley glacier at the seasonal and hourly scales (Gabbud et al., 2015); however, only the middle and lower elevations were detected as the glacier is relatively big. Another study reports the performance of the Riegl VZ®-6000 in monitoring the mass balance of five glaciers in the European Alps; the surface terrain of each glacier can be almost entirely detected using one scan position since these glaciers are very small and have steep terrain (Fischer et al., 2016). For medium-sized and large glaciers with flat terrain, however, one scan position cannot survey the whole glacier surface.

Urumqi Glacier No.1 has the most detailed annual and seasonal surface mass balance measurements in China. It is also one of the reference glaciers in the World Glacier Monitoring Service (WGMS) network due to its long data series, important location and significant local water supply (Li et al., 2011; Zemp et al., 2009). TLS surveys of Urumqi Glacier No.1 were initiated on 25 April 2015 for four scan positions (Fig. 1a), and the subsequent surveys correspond to measurement dates for glaciological mass-balance measurements. Multi-temporal high-resolution and high-precision TLS-derived DEMs are therefore available. To date, comparison of glaciological and geodetic mass balances of the glacier was reported for the period 1981-2009 at intervals of several years (Wang et al., 2014) and for the period 1981-2015 (Xu et al., 2018), but these studies used a series of low-quality topographic maps to calculate sub-decadal and decadal geodetic results. An accurate reanalysis of seasonal and annual glaciological mass balance of

Urumqi Glacier No.1 using high-resolution and high-precision DEMs has not been performed. Our previous study has used the TLS to implement two measurements one month apart (25 April-28 May 2015) to get monthly net mass balance of Urumqi Glacier No.1, whereas we simply compared glaciological and TLS-derived geodetic elevation changes of individual stakes, whether agreement between the glaciological and TLS-derived glacier-wide mass balance was pending, potential and shortcomes of such technology applied in seasonal and annual glacier mass-balance measurements in western China had not been discussed; besides we only considered snow/firn densities in the determination of a density conversion, which was used to convert monthly volume change to geodetic mass changes, as an abundance of fresh snow covered the entire glacier surface at the time of the TLS surveys (Xu et al., 2017). In fact, the volume-to-mass conversion becomes more challenging over short time periods as meteorological factors change mass balance gradients (Huss, 2013). Several recent studies have used an area-weighting method to calculate the annual density conversion by classifying a glacier surface into bare ice and firn (Fischer et al., 2016; Klug et al., 2018). But the volume changes in ice and firn/snow usually take place at the same vertical layer for summer-accumulation-type glaciers (accumulation and ablation take part simultaneously in summer months) from our field observations, it is therefore inappropriate for this study to adopt the area-weighting method. Besides, compaction and metamorphosis imply a shift in the vertical firn profile as well as changes in firn thickness and density (Cuffey and Paterson, 2010; Ligtenberg et al., 2011), so assuming no change occurs in the vertical firn density profile over time in the accumulation area is unrealistic (Bader, 1954).

This study takes Urumqi Glacier No.1 as an example and describes the use of the TLS to monitor annual and seasonal geodetic mass balances for two consecutive mass balance years (2015-17). The aim of this study is thus to established an optimization scheme of volume-to-mass conversion to realize the calculation of TLS-derived geodetic mass changes, to investigate the possible causes of the differences between glaciological and geodetic mass balance. The potential of such long-range TLS to measure mass balance of glaciers in western China is evaluated and several main considerations for a wide application of the TLS in glaciology are suggested.

## 2 Study site

Urumqi Glacier No.1 is a northeast-orientated small valley glacier, situated on the northern slope of Tianger Summit II (4848 m a.s.l.) in the eastern Tien Shan (43°06′ N, 86°49′ E, Fig. 1). This glacier covered a total area of 1.555 km$^2$ on 2 September 2015 according to TLS-derived high-resolution DEMs. Intensive glaciological investigations of Urumqi Glacier No.1 were implemented in 1959 and then a monitoring station (Tien Shan Glaciological Station) was set up for long-term glaciological mass-balance measurements. During the period 1959-2008, this glacier had experienced two accelerated mass loss, commencing in 1985 and 1996 respectively (Li et al., 2011). The glacier separated into two branches in 1993: the east branch and the west branch of Urumqi Glacier No.1.

Urumqi Glacier No.1 is a typical summer-accumulation-type glacier in a continental climate setting (Liu and Han, 1992; Li et al., 2011). The westerly circulation is influenced by the dynamic action of the Tibetan Plateau in the winter months, causing a cold climate with little precipitation in the study site (Han et al., 2006; Huintjes et al., 2010). During the summer month, the Tibetan Plateau becomes a thermal depression and forms a plateau monsoon, which carries warm and humid air from the India Ocean, producing abundant precipitation surrounding the Plateau (Huintjes et al., 2010). These climatic conditions were confirmed by the annual climate records (1959-2015) of Daxigou Meteorological Station, located about 3 km southeast of Urumqi Glacier No.1 at 3539 m a.s.l.; the annual average air temperature was about -5.0 ℃, and the annual average precipitation was 460 mm. 78% of the

annual total precipitation amount occurs from May to August (summer), dominated by solid precipitation (Yue et al., 2017). The climatic conditions determine that the glacier is dominated by weak accumulation from October to March (winter) and the accumulation is higher from April to May; both strong ablation and accumulation mainly take place between June and September (Liu et al., 1997).

## 3 Data and methodology

### 3.1 Terrestrial laser scanning

### 3.1.1 Principles and key features of Riegl VZ®-6000 TLS

Riegl VZ®-6000 TLS is an active laser imaging technique that calculates the distance between the object and the laser transmitter based on time-of-flight measurement with echo digitization and online waveform processing, and consequently the position of the point of interest to be computed (RIEGL Laser Measurement Systems, 2013). The scan mechanism includes a fast-rotating (60-120 ° from zenith) and more slowly rotating optical head (0-360 °). The mirror deflects the laser beam in different directions, thus forming a scan line from consecutive measurements. Meanwhile, the optical head rotates and this scan movement is called a frame scan. A line scan and frame scan generate a view scan using this technique; data collection occurs at a rate of 23 000-222 000 points per second and generates point clouds accordingly (RIEGL Laser Measurement Systems, 2014a).

The high-accuracy and high-precision ranging is based on its unique V-line technology of echo digitization and online waveform processing, which allows Riegl VZ®-6000 TLS to operate even in poor visibility and in demanding multi-target situations caused by dust, haze, rain, snow, etc. (RIEGL Laser Measurement Systems, 2014a).

### 3.1.2 Terrestrial laser scanning surveys

Multi-temporal terrestrial laser scanning data of Urumqi Glacier No.1 were collected from four scan positions to achieve maximum coverage, and each scan location was selected from the directions where most glacier surface point clouds would be achieved (i.e. the best possible visibility to glacier surface terrain) (Fig. 1c, d). To avoid ground motion and to obtain accurate coordinates of point clouds, each scan position was fixed using reinforced concrete with a standard GNSS-leveling point. The four scan positions were surveyed using the real-time kinematic (RTK) global navigation satellite system (GNSS, Unistrong E650 instrument) to give the most accurate direct georeferencing and registration. The 3-D coordinates were acquired in the UTM 45N coordinate system in the WGS84 datum. The accuracy of this type of survey has been reported to be within ±1 cm horizontally and ±2 cm vertically (Xu et al., 2017).

After the measurements of 3-D coordinates, the Riegl VZ®-6000 was mounted on a tripod placed in the scan position to survey the glacier surface terrain. The scan parameters and atmospheric conditions are of crucial importance, which directly determined point cloud data quality (point density and coverage) and acquisition time. The laser pulse repetition rate was first set to 50 kHz, then line resolution and frame angle measurement resolution were set to 0.2 ° to allow a view scan with vertical and horizontal angles were in the range of 60–120 ° from zenith and 0–360 °, respectively. A fine scan is a rectangular field-of-view scan, and the selected field should always cover the entire glacier to guarantee the overlap percentage of four scans was at least 30% (CH/Z 3017-2015, 2015). With each scan, the laser pulse repetition rate was reset to 30 kHz, and the corresponding line and frame resolution were

configured as 0.02 ° to ensure dense points of the glacier surface, except for the scan campaign on 2 September 2015 (Table 1). All scans are performed on sunny days (dry and windless atmosphere) to avoid the influence of precipitation and fog, which can absorb laser pulse and reduce the possible survey distance. Details of the survey parameters are listed in Table 1.

### 3.1.3 Point cloud processing

5   Raw data were post-processed with RiSCAN PRO® v 1.81; this includes direct georeferencing, data registration, vacuation and filtering (RIEGL Laser Measurement Systems, 2014b). For all five scan campaigns, four scan positions were used (Table 1). At first (i.e. direct georeferencing), the TLS data from the different scan positions had to be transformed from the Scanner's Own Coordinate System (SOCS) into a Global Coordinate System (GLCS). The transformation of a point from SOCS into the GLCS was described by Lichti et al (2005) and can be expressed by the following vector equation:

$$\vec{r_\mathrm{g}} = \vec{r_0} + R(k)\vec{r_\mathrm{s}} \,, \tag{1}$$

where $\vec{r_\mathrm{s}}$ is the vector of a target in the SOCS; $r_\mathrm{g}$ is the vector of the georeferenced target in the GLCS; $\vec{r_0}$ is the vector of SOCS origin in the GLCS, $k$ is the derived azimuth from the scan position to the backsight station and

$$R(k) = \begin{Bmatrix} \cos k & \sin k & 0 \\ \sin k & \cos k & 0 \\ 0 & 0 & 1 \end{Bmatrix}. \tag{2}$$

Hence the direct method of georeferencing uses the 3-D coordinates of the scan positions to realize its functions (Lichti et al., 2005;
Mukupa et al., 2016; Fey and Wichmann, 2017). The accuracy of the method depends on the quality of the measured coordinates. Previous studies stated that the direct georeferencing technique in TLS using GNSS is advantageous compared with total stations and the inclination sensors (Paffenholz et al., 2010; Mukupa et al., 2016).

The location of each scan was fixed in the GLCS after direct georeferencing; but the point clouds of the overlapped areas cannot
coincide completely due to the influence of orientation. In the second step, multi-station adjustment (MSA) was used for the data registration of each scan position according to the iterative closest point algorithm (Besl and McKay, 1992; Zhang, 1992). When we used MSA, the location of each scan was locked and the orientation of each scan was constantly adjusted in several iterations to compute the best overall fit for them based on least-squares minimization of residuals.

Afterwards we combined the overlapped scans in one layer. An octree algorithm was used to the merged layer to produce points with equal spacing to realize point cloud data vacuation (Schnabel and Klein, 2006; Perroy et al., 2010). A terrain filter was then applied to filter out noise and non-ground data due to atmospheric reflections such as dust or moisture, which still occurred despite scanning on fine days (RIEGL Laser Measurement Systems, 2014b). Finally, visual interpretation was performed to check the data and remove clear visual outliers,, and then glacier surface point clouds with one layer were produced.

## 3.2 Geodetic method

### 3.2.1 Geodetic mass balance calculations

As the orientation of each scan was continually adjusted to compute the best fit, the attitude angles of each scan campaign are different. Multi-temporal registration, also called relative registration, set the processed layer of 2 September 2015 as a reference; alignment of other scan campaigns onto the reference layer was finished with iterative closest point algorithms to determine the
spatial bias of the multi-temporal scans and extract accurate elevation changes (Revuelto et al., 2014; Gabbud et al., 2015). The relative registered layers were then exported into LAS data format for further processing. Multi-temporal registration of two

consecutive campaigns is a crucial step and determines the reliability of TLS-derived surface elevation changes (Revuelto et al., 2014; López-Moreno et al., 2016; Fey and Wichmann, 2017).

After the relative registration procedure, interpolation of the processed point cloud data calculated high-resolution DEMs of the study site. The surface elevation change $\Delta h_i$ at the individual pixel $r$ was calculated by differencing the TLS-derived multi-temporal DEMs with ArcMAP 10.2. The total volume change $\Delta V$ was determined by summing the elevation change $\Delta h_i$ of different time periods, and is expressed as

$$\Delta V = r^2 \sum_{i=1}^{N} \Delta h_i,$$ (3)

where $N$ is the number of total pixels covering the maximum extent of Urumqi Glacier No.1, and $r$ is the pixel size (1 m $\times$ 1 m).

The calculated volume change is converted to geodetic mass balance (m water equivalent (w.e.)) following:

$$B_{\text{geod}} = \frac{\Delta V}{\bar{S}} \cdot \frac{\rho}{\rho_{\text{water}}} = \frac{\Delta V}{1/2 \cdot (S_{t0} + S_{t1})} \cdot \frac{\rho}{\rho_{\text{water}}},$$ (4)

where $\bar{S}$ is the mean glacier area of the two acquisition dates $t0$ and $t1$, thinking a linear change over time, $\rho_{\text{water}}$ is the density of water and $\rho$ is the average bulk density (density conversion) of glacier volume change (Thibert et al., 2008; Zemp et al., 2013).

### 3.2.2 Density conversion

As described above, the geodetic mass balance is calculated based on volume changes, which require a density conversion. However, the density is difficult to determine; in most studies, it is estimated and not measured. Some researchers assume that no change occurs in the vertical firn density profile over time in the accumulation area and use glacier ice density for the conversion based on Sorge's law (Bader, 1954). Actually, the firn line, firn thickness and firn density all vary, and using the ice density causes an overestimate of mass balance. Huss (2013) recommended a density conversion of 850 $\pm$ 60 kg m$^{-3}$ for the volume-to-mass conversion based on an empirical firn densification model with idealized surface mass balance forcing. But the recommendation is appropriate in the case of a geodetic observation span longer than 5 years; with stable mass balance gradients, volume changes significantly different from zero and a firn area exists. Therefore, several recent studies classify the glacier surface into firn and bare-ice zones and use the volume-weighting method to calculate the annual conversion (e.g. Fischer et al., 2016; Klug et al., 2018). However, mass balance processes of Urumqi Glacier No.1 primarily occur in summer and the glacier is dominated by weak accumulation in winter (Liu et al., 1997). Glacier volume changes in ice and firn usually occur at the same vertical profile according to long-term observations; the surface classification is not applicable in this study. Here we use in situ measured thickness and densities of firn/snow ($\rho_{\text{firn}}$) and ablation stake data (change in ice thickness) to calculate single-point density conversion according to principle of glaciological mass balance calculations of Urumqi Glacier No.1 (Xie and Liu, 2010):

$$\rho_i = \frac{\Delta h_{\text{ice}} \cdot \rho_{\text{ice}} + \Delta h_{\text{firn}} \cdot \rho_{\text{firn}}}{\Delta h_{\text{ice}} + \Delta h_{\text{firn}}},$$ (5)

where $\rho_{\text{ice}} = 900$ kg m$^{-3}$ is glacier ice density, $\rho_{\text{firn}}$ is the density of firn/snow, $\Delta h_{\text{ice}}$ and $\Delta h_{\text{firn}}$ are the changes in ice and firn/snow thickness, respectively, which determined from glaciological mass-balance measurements. We extrapolated single-point values to the glacier-wide densities to calculate average bulk density ($\rho$) using the interpolation method (Table 3), and the distributed density conversions of the total glacier were then generated (Fig. 6a, c, e, g).

It is generally true that the density conversion relies on measurements of changes in the firn body, thickness and density of each firn layer being continuous from the top to the bottom of the snow pit, and a stratigraphic description of the firn layers is completed by experienced investigators. Major change processes in the snowpack (e.g. from crystals to grains, descriptive free water content

and ice layers, etc.) can be considered in this case (Kaser et al., 2003), and firn compaction assumed to be negligible. We use the volume-weighting method (the weights are the thickness changes of each firn layer and glacier ice) to calculate the firn density ($\rho_{\mathrm{firn}}$) of each snow pit in this study.

## 3.3 Glaciological method

### 3.3.1 In situ measurements

The mass balance of Urumqi Glacier No.1 has been observed using stakes or snow pits, since 1959 (Xie and Liu, 2010). Glaciological measurements broke off during the period 1967-79 and the glaciological data series during this period were reconstructed from correlations with climatic data observed at Daxigou Meteorological Station (Li et al., 2011). The program was re-established using glaciological methods in 1980. No less than 40 ablation stakes were drilled into the glacier and evenly distributed at different elevation bands using a stream drill, despite the fact that the number and location of stakes has varied from year to year, and snow pits were dug in the accumulation area (Fig. 1a). The mass-balance year of Urumqi Glacier No.1is defined from previous September 1 to next August 31 (Liu et al., 1997). Usually, from the beginning of May to early September each year, a spatial distribution of single-point ablation (mass loss) or accumulation (mass gain), and snow density (if there is snow cover) were measured by stakes and snow pits at monthly intervals. The net accumulation is measured by digging snow pits at each of the stakes in the area of the glacier where snow has accumulated during the period of investigation; stakes are drilled into the glacier and change in an exposed stake height plus change in snow depth (if snow exists) at two successive dates gives the net ice ablation at this point (Kaser et al., 2003; Xie and Liu, 2010). Hence the measured items include the stake vertical height over the glacier surface, thickness of superimposed ice, and the thickness and density of each snow/firn layer at individual snow pits. Note that fresh snow covered the whole glacier surface at the beginning of the ablation season, so snow pits must also be dug at each of the stakes.

### 3.3.2 Glaciological surface mass-balance determination

Glaciological mass balance includes point and glacier-wide mass balances. The rate of mass gain and loss per unit time is accumulation rate $\dot{c}$ and ablation rate $\dot{a}$, respectively, and the difference between these two, i.e. $\dot{c}$ minus $\dot{a}$ equals mass-balance rate $\dot{b}$. Integrating $\dot{b}$ over the time span from $t_0$ to $t_1$ gives point mass balance $\Delta b$ (Cogley et al., 2011):

$$\Delta b = \int_{t_0}^{t_1} \dot{b}(t)\, dt = b(t_1) - b(t_0). \tag{6}$$

Point values can be extrapolated to glacier-wide mass balance using the contour-line or profile method (Østrem and Brugman, 1991; Kaser et al., 2003). Here the time span is often a year or a season, and a seasonal mass balance is classically a winter balance or a summer balance (Cogley et al., 2011). Here $t_0$ and $t_1$ are the same as the $t$ defined by Eq. (4). Point values can be extrapolated to glacier-wide specific mass balance using the contour-line or profile method (Østrem and Brugman, 1991; Kaser et al., 2003). For Urumqi Glacier No.1, contour-line and isoline methods had successfully been used to calculate seasonal and annual glacier-wide mass balance (Xie and Liu, 1991), together with simulated values obtained using a simple energy-balance model (the energy divide into shortwave radiation and temperature dependent energy budget) in areas with no measurements (Oerlemans, 2010; WGMS, 2017). This study involves the glaciological measured data over the period 2015-17.

## 3.4 Accurate glacier boundary delineation

An accurate and updated glacier area was important for both geodetic and glaciological mass balance calculations (Zemp et al., 2013). Fresh snow cover probably led to an overestimate of glacier extent at the beginning of the ablation season. To reduce the

influence of snow cover and to extract accurate glacier outlines, we mainly considered glacier extents at the end of the ablation season. Glacier boundary delineation was performed following Abermann et al. (2010). Firstly three shade reliefs at the end of hydrological years 2015, 2016 and 2017 with an azimuth angle for illumination (300 °) were calculated based on multi-temporal high-resolution DEMs to show optimal visualization of contrasts in different aspects. We then delineated the glacier boundary
directly by manually digitizing the strongest roughness in the shade reliefs (Fig. 3). Areas of Urumqi Glacier No.1 were 1.555 km$^2$ in 2015, 1.550 km$^2$ in 2016 and 1.542 km$^2$ in 2017. Glacier area reduction was primarily attributed to terminus retreat (Fig. 2b, c).

## 4 Uncertainty assessment

### 4.1 Uncertainties of geodetic mass balance

After multi-temporal registration, errors related to the spatial bias of the multi-temporal DEMs may be negligible. Besides density
conversion for converting TLS-derived glacier surface elevation changes to mass balance, uncertainties in the geodetic mass balances derived from the TLS may be related to (1) errors in point cloud data acquisition, including surface terrain and atmospheric conditions (moisture and wind) (Revuelto et al., 2014; Fischer et al., 2016); and (2) errors in data processing and DEM generation, e.g. registration (multi-station adjustment), point cloud vacuation and filtering (smoothing terrain information) (Wheaton et al., 2010; Gabbud et al., 2015; Hartzell et al., 2015).

As mentioned in Sect. 3.1.2, dry and windless days were selected to perform the five scan campaigns. Instability of the TLS influences the registration of single scan positions from each data acquisition campaign, which includes small displacements of scan positions and the vibration of TLS. Each scan position was established on stable rock surfaces using reinforced concrete (the average drilling depth was greater than 80 cm) with a standardized GNSS-leveling point to avoid ground motion. In fieldwork,
TLS is mounted using a tribrach on a tripod to level the instrument (Xu et al., 2017). Revuelto et al. (2014) found that the vibration of TLS can introduce considerable errors in measurements performed over large scales. In our experience, this issue is mainly relevant to wind, so windless weather conditions are important. Because the registration error cannot be distinguished from the positional uncertainties and the surface, it is difficult to assess registration-induced uncertainty; the error statistics are usually used to evaluate the registration error (Fey and Wichmann, 2017). RiSCAN PRO® v 1.81 software reports error statistics of the MSA
results (RIEGL Laser Measurement Systems, 2014b). The standard deviation of errors ($\sigma_{MSA}$) from the set of residuals obtained from registering the point cloud can be considered as an indication of registration quality (Gabbud et al., 2015; Fischer et al., 2016). Values of $\sigma_{MSA}$ over stable terrain surrounding Urumqi Glacier No.1 are listed in Table 2 for the four periods. Registration quality was higher at seasonal than at annual scales; the higher quality may be attributable to fresh snow cover, which makes the stable terrain smooth.

Despite four scan positions placed at the terminus of Urumqi Glacier No.1, two undetected areas of west branch exist (two green polygons in Fig. 5) due to flat terrain ranges from 4050 to 4100 m a.s.l. (Fig. 1a), where the emission laser cannot be received by the laser receiver. We filled these regions using the spatial interpolation method, which can induce potential errors in DEM creations. The lack of dense measured 3-D coordinates of the terrain limits us to assessing terrain-induced errors qualitatively. For
precision, the undetected areas were not taken into account in calculating the geodetic mass balance in fact, related errors were small as the relative proportions of the two areas over the entire glacier surface were minor (3.1% for summer 2015, 3.2% for 2015-16, 3.6% for summer 2016, and 4.6% for 2016-17, Fig. 5). Furthermore, supraglacial river exists at the strong ablation season

(June to September due to glacier melting (Fig. 1c, d), which was detected by the TLS surveys. In order to preserve terrain information as much as possible, the octree algorithm built the topological relationship of scattered points to realize the vacuation of the point cloud. Point cloud filtering is also a significant post-processing step because of the dense ablation stake network, which is actually scanned by the device. Fortunately, fine scan generates high-density points of the glacier surface terrain.

No better ways can be used to evaluate the uncertainty of DEMs without precise and well-distributed stable points (Bolch et al., 2017). The standard error ($\sigma_{\overline{\Delta hTLS}}$) of elevation changes over stable terrain can be considered as a criterion of the uncertainty of the entire glacier (Rolstad et al., 2009; Zemp et al., 2013). The standard deviation ($\sigma_{\Delta hTLS}$) of the stable terrain elevation changes is suitable for estimating the uncertainty of the DEM differences at the individual pixel scale (Fig. 3); in this case the standard error is

10 defined as the standard deviation. However, the spatial auto-correlation must be considered when we calculate the uncertainty of the glacier-wide elevation changes. Thereby, the uncertainty of TLS-derived glacier-wide elevation changes ($\sigma_{\overline{\Delta hTLS}}$) for individual glaciers were quantified using the geostatistical analysis methods of Rolstad et al. (2009) and written as

$$\sigma^2_{\overline{\Delta hTLS}} = \sigma^2_{\Delta hTLS} \cdot \frac{1}{5} \cdot \frac{S_{cor}}{S}, \tag{7}$$

where $\sigma_{\Delta hTLS}$ denotes the standard deviation of TLS-derived elevation changes over stable terrain. $S_{cor}$ is spatially correlated area.

Given the high density ($> 1$ point m$^{-2}$) of the TLS data, we can probably assume that the number of independent items is about the number of glacier pixels (cf. Joerg et al., 2012). Here we therefore assume $S_{cor} = S$. This leads to calculated values of $\sigma_{\overline{\Delta hTLS}}$ range from $\pm0.16$ to $\pm0.25$ m (Table 2).

Uncertainties related to the density conversion for a single point ($\sigma_{\rho i}$) were calculated as

$$\sigma_{\rho i} = \frac{\Delta h_{ice} \cdot \sigma_{\rho ice} + \Delta h_{firn} \cdot \sigma_{\rho firn}}{\Delta h_{ice} + \Delta h_{firn}}, \tag{8}$$

where $\sigma_{\rho ice}$ and $\sigma_{firn}$ are uncertainties of ice and firn densities, which were assumed to be $\pm17$ and $\pm50$ kg m$^{-3}$, respectively, following Klug et al. (2018). We then extrapolated single-point values to glacier-wide uncertainties ($\sigma_\rho$) using the interpolation method on the ArcMAP 10.2 platform (Table 3). According to Huss et al. (2009), the uncertainties of the geodetic mass balance ($\sigma_{geod}$) can be estimated using

$$\sigma_{geod} = \pm\sqrt{\left(\overline{\Delta hTLS} \cdot \sigma_\rho\right)^2 + (\rho \cdot \sigma_{\overline{\Delta hTLS}})^2}, \tag{9}$$

where $\overline{\Delta hTLS}$ is the average of TLS-derived glacier-wide elevation changes and the related uncertainty relies on the accuracy of the used DEMs.

## 4.2 Uncertainties of glaciological measurements

There are additional sources of error in the glaciological measurements that lead to uncertainties in glaciological mass balance that are not easy to quantify (Dyurgerov, 2002). These uncertainties were classified into three groups: (i) errors in field observations, (ii)

errors related to spatial extrapolation over the entire glacier and (iii) errors due to non-updated glacier area. Note that the class (iii) uncertainties appeared to be negligible due to the short time intervals (two consecutive years) in our study.

Point measurement uncertainties are prone to errors in stake readings and snow/firn density measurements (Jansson and Pettersson, 2007; Thibert et al., 2008; Huss et al., 2009), sinking or melting-out of stakes and misidentification of the firn layer surface at the

35 end of the last hydrological year (Zemp et al., 2010). Huss et al. (2009) demonstrated errors of $\pm0.1$ and $\pm0.3$ m w.e. for reading stakes in the ablation and accumulation areas, respectively. Zemp et al. (2010) determined an overall stochastic uncertainty at $\pm0.2$

m w.e. a$^{-1}$ for field measurements. Zemp et al. (2013) reanalyzed the mass balance of Hintereisferner, Austria, from 1953 to 2006 (six time intervals) and found an uncertainty of ±0.2 m w.e. a$^{-1}$ for point mass balance. Beedle et al. (2014) suggested an error of point mass balance to be about ±0.1 m w.e. a$^{-1}$ for accumulation-area measurements. For Nigardsbreen (Norway) glacier, Andreassen et al. (2016) calculated a point measurement of ±0.25 m w.e. a$^{-1}$ by summing false determination of the summer surface (±15 m w.e. a$^{-1}$), subsidence of stakes (0.20 m w.e. a$^{-1}$), errors in snow (0.05 m w.e. a$^{-1}$) and firn (0.02 m w.e. a$^{-1}$) density measurements. Following Thibert et al. (2008), here errors of ablation measured in ice ($\sigma_a^{ice}$) and firn ($\sigma_a^{firn}$) are calculated using $0.14/\sqrt{N_a^{ice}}$ and $0.27/\sqrt{N_a^{firn}}$, respectively, where $N_a^{ice}$ and $N_a^{firn}$ denote the number of ablation stakes and snow pits (if firn exists), respectively; errors in accumulation measurements ($\sigma_c$) are determined based on $0.21/\sqrt{N_c}$, where $N_c$ is the number of snow pits.

The class (ii) errors originate from extrapolating observed values to unmeasured areas, insufficient spatial distribution of measured sites and the interpolation method. Hock and Jensen (1999) evaluated the error of the interpolation method at about ±0.1 m w.e. a$^{-1}$ for mean specific mass balances. Huss et al. (2009) computed and compared mean specific net balance with randomly reduced annual stake datasets and found that the error was ±0.12 m w.e. a$^{-1}$. For Urumqi Glacier No.1, we find that the differences between specific net mass balance at individual sites and in situ measured point mass balance at corresponding sites were in the range of 0-0.04 m w.e. with an average value of 0.01 m w.e., namely, the error of spatial interpolation in the measured area is small. Firn basin and glacier tongue terrain of the west branch are very steep and the upper eastern elevation of the east branch is also precipitous, resulting in no in situ measurements are available in theses inaccessible areas. Therefore the error mainly originates from unmeasured areas (e.g. accumulation areas) but the lack of measured data in the accumulation areas limits us to quantify relevant uncertainties. We conservatively assume that the corresponding uncertainty $\sigma_{extra}$ was ±0.1 m w.e. a$^{-1}$ (cf. Andreassen et al., 2016).

Taking into account the above-mentioned factors, the uncertainty of the glaciological mass balance $\sigma_{glac}$ is calculated as

$$\sigma_{glac} = \sqrt{(\sigma_a^{ice})^2 + (\sigma_a^{firn})^2 + \sigma_c^2 + \sigma_{extra}^2} \; . \tag{10}$$

Resulting values of $\sigma_{glac}$ are listed in Table 3.

## 5 Results

### 5.1 Spatial patterns in TLS-derived surface elevation changes

The high-accuracy and -resolution DEMs allowed a detailed insight into the glacier surface elevation changes. Distributed elevation change patterns are generally similar for the four periods, i.e. both branches are characterized by a lower-elevation thinning of 1-3.5 m, elevation changes are more positive and show smaller lowering to pronounced thickening in the upper-elevation parts except for west branch in the mass-balance year 2016-17 (Fig. 4a, c, e), this altitudinal change patterns are in good agreement with the long-term glaciological measurements.

Compared to the mass-balance year 2015-16, areas of clearer increase were observed in the upper eastern parts of east branch in the mass-balance year 2016-17, but ice losses in the lower-elevation parts and glacier thickening in the upper reaches of west branch were greater in the previous mass balance year (Fig. 5c, g). Surface lowering in summer 2015 mainly occurred in the ablation areas of east branch (Fig. 4a), and glacier surface ablation was significantly greater in summer 2016 than in the first summer (Fig. 4e). For a completed mass balance year 2015-16, glacier thinning areas and values in summer were obviously bigger than the whole year, which may be related to fresh snow covered the glacier at the beginning of ablation season. In addition, there were some

curves of pronounced glacier surface lowering in the ablation areas during summer periods, which were related to supraglacial river (Fig. 1c, d). An area of minor thinning is detected at the lower lift (northerly) edge of east branch, which may be associated with debris cover (Fig. 1c).

## 5.2 Glacier-wide elevation and geodetic mass changes

TLS-derived glacier-wide mass balances (Table 3) and their spatial distributions (Fig. 5b, d, f, h) were calculated by multiplying the spatially distributed glacier surface elevation changes (Fig. 4a, c, e, g) with the corresponding distributed density conversion (Fig. 5a, c, e, g). The thicker snow and firn covered the whole glacier surface at the beginning of May each year and the ablation area was bare ice or covered by a thin snow layer at the end of the ablation season according to field observations (Liu et al., 1997; Xie and Liu, 2010), so the changes in ice and firn/snow thickness are observed during the summer months. However, firn and snow

densities are far smaller than glacier ice density, these result in annual single-point density conversion $\rho_i$ is bigger and the glacier-wide annual density conversions were accordingly higher than the summer ones (Table 3). From the density conversions, we can conclude that the summer geodetic mass balance was highly affected by the snow and firn, and the magnitude of the altitudinal variability in the summer mass balance significantly changed when compared to elevation changes (Fig. 4a, c), whereas the spatial distributed patterns between elevation changes and mass balances showed good performances. These suggest that the density

conversions vary for all of the studied periods and a constant value used as the conversion is clearly inappropriate. Here in situ measured densities from snow pits improved the accuracy of TLS-derived geodetic mass-balance calculations and therefore provided exceptional level of detail on glacier-wide mass balance.

Urumqi Glacier No.1 experienced negative surface elevation changes and mass balances for all of the four investigated periods

(Table 3). Summer elevation lowering and mass loss were slightly greater than annual decreases, which may be related to the climatic conditions observed at Daxigou Meteorological Station (see discussion in Sect. 2). In the mass-balance year 2015-16, calculated glacier-wide geodetic mass balance was -0.72 ±0.17 m w.e., which was slightly more negative than in the second mass-balance year. Summer and annual mass balances of west branch were more negative compared to the corresponding values of east branch, except for summer 2016 when the mass loss of east branch was greatest.

## 5.3 Comparison to in situ glaciological measurements

TLS-derived geodetic elevation changes at individual stakes closely matched the glaciological elevation change (changes in stake height) of individual stakes from in situ measurements and the difference ($\Delta h_{TLS}$ -$\Delta h_{glac}$) in surface elevation changes were close to zero for most of the point measurements (Fig. 4a, c, e, g). The correlation coefficients ($R^2$) between the glaciological elevation change at the ablation stakes and TLS-derived geodetic elevation change at corresponding points were more than 0.90 (Fig. 4b, d, f,

30   h). Note that the location and number of ablation stakes varied slightly over time due to stake melt-out, thus we selected the best-monitored single-point results to objectively assess the accuracy of the geodetic elevation changes. The majority of the point elevation changes from TLS measurements were slightly positive compared to the glaciological ones, except for summer 2016. During summer 2016 and mass-balance years 2015-16 and 2016-17, the mean values of $\Delta h_{TLS}$ - $\Delta h_{glac}$ were 0.18, 0.25 and 0.14 m, respectively, which are systematically less than the corresponding uncertainties ($\sigma_{\overline{\Delta h TLS}}$) of the glacier-wide elevation differences.

The varying tendencies of the glaciological mass balances coincided with the geodetic ones (Table 3). In 2016-17, the difference ($\Delta B = B_{glac} - B_{geod}$) in glacier-wide mass balances of Urumqi Glacier No.1 between the glaciological and geodetic methods was close to zero. Significant differences between the two methods were detected in summer 2015 for Urumqi Glacier No.1 and east

branch, with $\Delta B$ = -0.24 m w.e. and $\Delta B$ = -0.27 m w.e., respectively. In other three periods, the differences were much less the uncertainties of $\Delta B$, which were calculated based on the law of error propagation ($\pm \sqrt{\sigma_{\text{geod}}^2 + \sigma_{\text{glac}}^2}$). Overall, the differences of Urumqi Glacier No.1 were slightly smaller than those of the two branches (Fig. 6). In order to calculate the statistical significance between the two methods and validate the geodetic against the glaciological mass balance, the reduced discrepancy ($\delta$) between the

two methods was calculated following Zemp et al. (2013, Eqs. 19-21); the results of $\delta$ range from -1.14 to 0.58. As $\delta$ falls within the 95% ($|\delta| < 1.96$) and 90% ($|\delta| < 1.64$) confidence interval, good agreement between the glaciological and geodetic methods can be considered as satisfactory.

Spatially distributed differences between glacier-wide glaciological and geodetic mass balances were calculated to give the spatial

deviations. Over most parts of the glacier surface, especially for the areas near the best-monitored points, deviations were small, indicating both methods showed very close spatial results. Pronounced differences mainly occurred on the steep slopes where in situ measurements were missing (Fig. 7a, d, g, j). The mass balance elevation distribution derived from the two methods remained similar despite the presence of differences in magnitude, i.e. mass balance increased with rising altitude (Fig. 7b, c, e, f, h, i, k, m). The geodetic results were more positive in lower-elevation regions and more negative in the higher glacier parts in general

compared with the glaciological mass balance, which were probably related to glacier dynamic processes (discussed in Sect. 6.4). The dotted (glaciological) and solid (geodetic) lines met where the glacier mass balances were close to zero; this meant that the equilibrium-line altitudes (ELAs) derived from the two methods matched closely, especially in mass-balance year 2015-16 and summer 2016 (Fig. 7e, f, h, i), but the biggest shift between the two methods was detected in summer 2015 for east branch, which may be related to survey data differences between the glaciological and geodetic observations (see details in Sect. 6.5). This

reflects that the TLS can be therefore considered as an effective tool to calculate ELA.

## 6 Discussion

### 6.1 The quality of point cloud data and DEM differencing

The important factors for scanning high-quality point cloud data are visual angles of the scan positions and atmospheric conditions. A dry and windless atmosphere is a prerequisite for high-quality data acquisition. Good visual angles can easily be achieved for

very small cirque glaciers. Generally, the area and length of reference glaciers are greater, with a huge variation in altitude. The maximum working distance (6 km) of Riegl VZ®-6000 is specified for flat targets with size in excess of the laser beam diameter, perpendicular angle of incidence, and atmospheric visibility in excess of 23 km. In bright sunlight the operational range may be considerably shorter than under an overcast sky (RIEGL Laser Measurement Systems, 2014a). However, glaciers generally have complicated surface terrain and the requirement of perpendicular angle of incidence is not always met, so the unscanned regions

usually have flat terrain (Fig. 1d). It is very difficult for us to get a dry and windless atmosphere under an overcast sky around a glacier. In these situations, more than one scan position must be set in order to scan as much of the glacier surface area as possible. However, this, in turn, can create errors in data registration. The average error originating in MSA ($\sigma_{\text{MSA}}$) of the investigated periods was ±0.16 m (Table 2). Actually, $\sigma_{\text{MSA}}$ is highly dependent on the overlap percentage of point clouds from each scan position of the same survey data and the accuracy of global 3-D coordinates of each scan position. In our fieldwork, the overlap

percentage of point clouds from the four scan positions was more than 30%, which met the requirements of data registration (CH/Z 3017-2015, 2015). Note that we did not get better results with more scan positions since this would probably decrease the quality of the MSA. We should find the best visual angles to obtain the maximum scan range with the fewest scan positions. Higher

elevation favors better angles, but it is not always easy for us to access higher parts and place the instrument. For the ideal distribution, scan positions should be located in different elevation bands and directions (Fig. 1a), and we can also mount a steadying bar on a tripod to raise the altitude of the TLS.

Systematic shifts of DEMs in the horizontal and vertical directions can also increase the uncertainty of DEM differencing (Nuth and Kääb, 2011), so multi-temporal registration of two consecutive scan campaigns is predicated on the TLS-derived geodetic elevation changes being accurate. The mean uncertainty of elevation changes was ±0.22 m (Table 2), which was slightly smaller than in the TLS datasets used for other glacier thinning measurements (e.g. López-Moreno et al., 2016). This may be attributable to the use of accurate global 3-D coordinates of each scan positions and a sufficient number of stable terrains (Fig. 3). In addition,
fixed scan positions also reduce multi-temporal registration error and enhance the accuracy of glacier-wide elevation changes. So the quality of point cloud data and DEM differencing is encouraging.

## 6.2 Accuracy of geodetic and glaciological mass balance

It is obvious that the quality of TLS-derived geodetic mass balances relies on the accuracy of glacier surface elevation changes and density conversion of volume to mass changes. With regard to density conversion, our approaches account for the changes in ice
and firn/snow thickness as well as the corresponding densities to calculate more accurate values of density (Table 3). The annual values for $\rho$ were in the range of 763-865 kg m$^{-3}$, which was in line with the average density of $850 \pm 60$ kg m$^{-3}$ recommended by Huss (2013), whereas, $\rho$ of west branch declined to 763 kg m$^{-3}$ in the mass-balance year 2015-16, largely because of the presence of fresh snow cover at the time of the glaciological measurements. For the same reason, the summer values concurred with the recommended density. Calculated uncertainties in the geodetic mass balances ranged from ±0.13 to ±0.20 m w.e., with an average
value of ±0.16 m w.e. for the investigated periods (Table 3) and were slightly greater than other geodetic mass balance measured base on the same TLS device (e.g. Fischer et al., 2016), which may be related to multiple scan positions and a larger scanning range, but smaller than those derived from remote sensing imagery (e.g. Holzer et al., 2015; Barandun et al., 2018).

Dense spatially measured sites cover the glacier surface (the average density is about 28 stakes km-2 from 2015 to 2017), except
for the inaccessible areas (Fig. 1a), to measure the glaciological mass balance. The mean uncertainty of $\sigma_{glac}$ was ±0.12 m w.e. and mainly originated from spatial extrapolation of point measurements. This value is smaller than most recent studies (e.g. Andreassen et al., 2016; Thomson et al., 2017; Klug et al., 2018). This is probably due to relative smaller area and accompanying higher density of point measurements of Urumqi Glacier No.1 than aforementioned studies (Fig. 1a). Thus the TLS device yields accurate geodetic results and the quality of the glaciological mass balances is also very good. Nonetheless, uncertainties in field
measurements and interpolation can potentially contribute the deviations between glaciological and geodetic mass balances. The obvious deviations in the spatial distributed differences were found at the firn basin of west branch and the upper right edge elevations of east branch, two unscanned areas also present big deviations (Fig. 7a, d, g, j). Remarkable differences of east branch in summer 2015 induce poorest match between glaciological and geodetic mass balance elevation distribution (Fig. 7c), this can be explained by the lack of well-measured point data (Fig. 7a).

## 6.3 The influence of internal and basal mass balances

The glaciological method cannot measure internal and basal mass balances, but these processes are implicitly captured by the repeated geodetic surveys. We need to provide a rough estimate of internal and basal mass balances of Urumqi Glacier No.1 to detect their contributions to the differences between glaciological and geodetic mass balances.

Urumqi Glacier No.1 is a small and cold glacier, basal sliding and bed deformation of the glacier are negligible since the temperature at the glacier bed is below the melting point of ice, so the glacier has low ice velocity and dynamics, and hardly any subglacial water systems (Huang, 1999; Xie and Liu, 2010; Wang et a., 2017). Previous studies have suggested that internal ablation of poly-thermal glaciers is negligible as the ice motion is small (e.g. Albrecht et al., 2000; Zemp et al., 2010). In addition,

internal melt caused by changes in potential energy due to glacier dynamics is negligible, as the glacier dynamics themselves are insignificant. Thus, internal ablation of Urumqi Glacier No.1 is weak, and mainly comes from the released potential energy of descending water:

$$B_{pe} = \frac{Q_m g}{L_f \bar{s} \rho_{water}} \cdot \frac{\bar{h}_{ELA} - h_{term}}{2}, \tag{11}$$

where $Q_m$ denotes annual discharge of flowing water, g is the gravitational acceleration, $L_f$ is the latent heat of fusion. Glacier mass

loss mainly comes from ablation area and we assume meltwater originates at half the vertical from the glacier terminus to the ELA, so $\bar{h}_{ELA}$ and $h_{term}$ are average equilibrium-line altitude (ELA) (4152 m) and the altitude of the glacier terminus (3775 m), respectively, $\bar{s}$ is the average glacier area between 2015 and 2017. The cumulative measured glacier surface ablation over the two years was used to determine annual discharge and the value of $Q_m$ was estimated to be about $1.4 \times 10^9$ kg a$^{-1}$. Equation (11) gives $B_{pe}$ = -0.005 m w.e. a$^{-1}$. Internal accumulation results from refreezing percolating water or the freezing of capillary-trapped water in

cold snow and firn, which plays a significant role in cold and continental climates, such as Storglaciären, Sweden (Zemp et al., 2010) and Castle Creek Glacier, Canada (Beedle et al., 2014), Urumqi Glacier No.1 is a cold glacier in a continental climate setting with no measurements of internal accumulation, so we conservatively estimate internal accumulation to be about 4% of the winter mass balance and the resulting value was about 0.01 m w.e. a$^{-1}$. Final calculated internal mass balance of Urumqi Glacier No.1 was about 0.005 m w.e. a$^{-1}$.

Basal ablation is generally attributed to frictional heat of basal sliding and geothermal heating for mountain glaciers (Thibert et al., 2008; Thomson et al., 2017; Galos et al., 2017). Basal ablation caused by frictional heat is very small since there is hardly any basal sliding of Urumqi Glacier No.1. Here we mainly consider the contribution of basal ablation from geothermal heat ($B_{gt}$), which was estimated using

$$B_{gt} = \frac{qt}{L_f \rho_{water}}, \tag{12}$$

where $q$ = 0.059 W m$^{-2}$ is the geothermal heat flux (Huang, 1999), $t$ is the mass-balance period; here we primarily consider annual scale and basal ablation was estimated to be about 0.005 m w.e. a$^{-1}$. The calculated internal and basal ablation totaled -0.01 m w.e. a$^{-1}$.

35 Finally the total value of internal and basal mass balances was close to zero, which is far less than the difference ($\Delta B$) between the two methods. This suggests that the contribution of annual internal and basal processes is negligible and does not affect the differences between the two methods.

## 6.4 Glacier vertical velocity component

Geodetic measurements of glacier surface elevation changes include glacier surface mass balance and vertical velocity component (Kaser et al., 2003; Geist et al., 2005). Vertical velocity ($w_s$) is downward (submergence) and causes dynamic thinning of the glacier in the accumulation area, and in the ablation area, vertical velocity is upward (emergence) and results in dynamic thickening of the glacier. This dynamic process results in the general difference between the elevation-distributed mass changes stated above (Fig. 8). To discuss the influences of vertical velocity component, here $w_s$ depends on the kinematic boundary condition at Urumqi Glacier No.1 surface as basal sliding and bed deformation of the glacier are negligible (cf. Petterson et al., 2007; Cuffey and Paterson 2010):

$$\dot{h} = \frac{\dot{b}}{\rho} + w_s - u_s \frac{\partial s}{\partial x} - v_s \frac{\partial s}{\partial y} \tag{13}$$

in which $\dot{h}$ is the rate of glacier surface elevation changes, $u_s$ and $v_s$ are the components of horizontal ice velocity at the glacier surface s, respectively (Cuffey and Paterson 2010). We neglect the advection of the glacier surface topography induced by horizontal ice flux due to the low reduced horizontal velocity (Wang et al., 2017) and short time intervals of this study. Then changes in $\dot{h}$ equals the sum of $\frac{\dot{b}}{\rho}$ and $w_s$ (Beedle et al., 2014), can be expressed as

$$\dot{h}\rho - \dot{b} = w_s\rho \tag{14}$$

Glacier dynamic thinning and thickening can be detected by subtracting the glaciological mass balances from the geodetic ones. For most of the study periods, positive difference values (thickening) dominate in the lower elevations, especially the glacier tongue, and negative difference values (thinning) mainly occur in the higher parts (Fig. 7a, d, g, j). Positive values across the east branch in summer 2015 may be related to different survey dates between the geodetic and glaciological methods.

Applying a reciprocal density conversion to the mass balance differences provides estimates of the submergence and emergence velocities. Here we defined the term submergence as negative vertical velocity and emergence as positive vertical velocity. Variation tendency of the estimated velocities at ablation stakes were found to match the in situ measured ones, especially for east branch (Fig. 8). Relative bigger differences of west branch were detected in the mass balance years 2016-17 (Fig. 8b, d), which may due to an avalanche in the upper part during the summer 2017. The firn basin terrain of west branch is very steep and is adverse to mass accumulation, which can also be validated in terms of TLS-derived glacier surface elevation changes (Fig. 4g). Thus pronounced misalignment of mass balance elevation distribution curves between the two methods occurred. Considering the errors of estimate and in situ measurements, submergence and emergence velocities can be estimated using the TLS-derived DEMs and glaciological mass balance. The difference in mass balance elevation distribution can be largely explained by glacier dynamic thinning at higher elevations and dynamic thickening at lower elevations.

In fact, the vertical velocity of Urumqi Glacier No.1 is small (Fig. 8), we now discuss the errors of glacier surface elevation changes versus dynamic thinning and thickening. Differences in glacier surface elevation changes derived from the TLS and glaciological measurements were close to zero for the vast majority of the ablation stakes, and corresponding errors in the differences were mostly larger than the difference themselves (Fig. 9). Compared with the errors of measurements, dynamic thinning and thickening of the glacier were minor and negligible. So Riegl VZ®-6000 TLS can be considered as an effective tool to measure the mass balance of Urumqi Glacier No.1

## 6.5 Meteorological and glacier surface terrain considerations

Although glacier thinning and thickening were negligible, other factors such as meteorological conditions and glacier surface terrain may cause mass balance differences between the two methods. Figure 9 shows daily meteorological records provided by Daxigou Meteorological Station from 25 April 2015 to 28 August 2017. Positive temperature and more than 75% of the annual total precipitation amount occurred during the summer months; this probably resulted in summer mass balances that were slightly more negative than annual ones (Table 3). Although the reduced discrepancies between TLS-derived geodetic and glaciological mass balances fall within the 95%, a bigger difference ($\Delta B$) between the two datasets is detectable for east branch, and the difference is caused by survey date differences. The glaciological measurements of east branch were performed five days before the TLS surveys (28 August-2 September 2015). The total precipitation was 67 mm and daily mean temperatures were close to zero, besides the daily minimum temperatures were all below 0 ℃ during the five days (Fig. 9). The climatic conditions may be responsible for the larger mass-balance differences of east branch. In addition, light snow occurred before the last TLS surveys (28 August 2017), possibly resulting in the noticeable increase in the upper eastern parts of east branch (Fig. 5h).

The differences in mass balance between the two methods were possibly related to the effect of glacier surface terrain. The presence of two minimal unscanned areas in TLS surveys is due to the flat terrain of west branch surface (two green polygons in Fig. 5). The geodetic mass-balance calculations did not include these unscanned areas; this cloud potentially increased the difference between the two methods. Furthermore, these undetected regions located in the ablation area and higher wastage than the surroundings were observed according to glaciological measurements. This may imply that the geodetic mass balances of west branch were more positive than the glaciological ones (Table 3) and a discrepancy in mass balance elevation distributions of west branch were observed at 4000-4150 m a.s.l. Nevertheless, the geodetic method is able to cover the majority of the glacier surface and take the terrain characteristics into account, whereas the glaciological measurements cannot capture all the topographic features despite a dense spatial coverage of in situ observations being applied, and what's more, in situ observations are missing in the firn basin and glacier tongue terrain of west branch and eastern elevations of east branch because of the presence of precipitous terrains in these inaccessible regions (green color in Fig. 11a). The eastern elevations of east branch are dominated by the northwest aspect, and the firn basin have aspects from north to northwest (Fig. 11b), aspects are likely to influence the glacier surface albedo and thereby control the surface change patterns (cf. Yue et al., 2017).

## 6.6 Potential of the long-range TLS applied in glacier mass balance monitoring

This study presents the application of multi-temporal Riegl VZ®-6000 TLS point clouds in mass balance monitoring of Urumqi Glacier No.1. The long-range TLS can provide high-temporal-spatial-resolution and -accuracy DEMs to allow more detailed insight into glacier evolution (e.g. Gabbud et al., 2015). To take advantage of this and provide more-precise glacier surface elevation changes, it is worth remembering that fixed scan positions are highly important between consecutive scans when using our approach. We should also note that not all glaciers in China are as easily accessible as Urumqi Glacier No.1. For many large glaciers, it is not always easy to fix scan positions using reinforced concrete with a standard GNSS-leveling point, but we can mark stable bedrock outcrop as a scan site. Another advantage of this type of TLS is the long scanning range, and such an instrument could allow most of the glacier surface to be scanned from one or several scan positions, especially for remote and inaccessible glacier areas (e.g. crevasses, steep ice, debris cover, etc.). Therefore the instrument provides a quantitative evolution in spatial coverage compared to glaciological in situ measurements, which can be seen as a beneficial complement to glaciological mass balance, particularly for calibrating inaccessible areas. TLS surveys can also provide updated glacier boundary and surface DEMs.

In addition, we can paste several retro-reflective targets (e.g. reflective foils, corner cube reflectors and retro-reflective paintings) to the surface of each stakes and the targets can be easily surveyed and identified since each of them has a high directivity of the reflected laser radiation, then the location of stakes can be determined. All of these parameters are favorable for glaciological mass-balance calculations. A combination of glaciological and TLS observations may yield optimum results. TLS-derived geodetic
results can validate the distributed glacier mass-balance models as the TLS can provide high spatial and temporal resolution measurements, especially in the strong ablation season, the instrument can be used to investigate daily or sub-daily ablation (e.g. Haut Glacier d'Arolla, Switzerland; Gabbud et al., 2015), which can completely meet the requirements of time resolution for glacier mass-balance models.

One drawback of the TLS surveys is the presence of data voids (unscanned areas), even for very small glaciers (e.g. Fischer et al., 2016). This is due to limited scanning angle and complex glacial terrain. An emerging low-cost Unmanned Aerial Vehicles (UAV) has the potential to avoid data voids in glaciological monitoring since the good surveying angle of UAV. Immerzeel et al. (2014) showed that UAV combined with a Structure from Motion (SFM) workflow provide a powerful tool for monitoring mass balance and surface velocity of a Himalayan glacier with high spatial accuracy. From our field experiment at Urumqi Glacier No.1, rarefied
air and frequent blustery wind around glaciers usually induce the power of UAV were nondurable, and rock outcrops results in difficult operations of such instrument. Hence we mainly consider using UAV to survey unscanned area, integrating of UAV- and TLS-acquired data can provide the whole glacier surface terrain of interest. Other technology such as terrestrial photogrammetry also has the ability to estimate mass balance, and the quality of photogrammetric estimation is similar to the quality of TLS (e.g. Piermattei et al., 2015; Fugazza et al., 2018). However, the reliable of UAV and terrestrial photogrammetry in glacial environments
is more dependent on the natural features (i.e. characteristic image objects) of the surveyed surfaces compared with TLS. The cost of TLS is higher than UAV and ground-based photogrammetric surveys.

From our experience, the monitoring tool is potentially applicable to other glaciers provided that these glaciers have small to medium size and relative steep terrain. According to the second Chinese glacier inventory (Guo et al., 2015), ~83% and ~70% of
the total number of glaciers in China have an area smaller than 1 $km^2$ and 0.5 $km^2$, respectively, and only ~3% with an area lager than 5 $km^2$. Riegl VZ®-6000 TLS has been proved successful in monitoring mass balance of very small glaciers (area smaller than 0.5 $km^2$) in the Swiss Alps (Fischer et al., 2016). Hence it should be possible to measure most glaciers using the TLS. Furthermore, if we assume that these glaciers with an area≤1.6 $km^2$ (approximate area of Urumqi Glacier No.1) and a mean surface slope greater than 23.4 ° (slope of Urumqi Glacier No.1) have a good visibility to be monitored using the TLS, the proportion of theoretical
appropriate glaciers is ~58.5% ,which evenly distributed at different mountains (Fig. 12).  However, it is not always easy for us to monitor all of the appropriate glaciers as some of them are located in remote areas (i.e. far away from road), now we have selected some benchmark glaciers with easily accessible locations for future application of TLS measurements, the TLS system thus has a huge potential for glacier mass-balance monitoring in China.

Nevertheless, TLS measurements and point cloud data post-processing are challenges for a broader application. One disadvantage of the TLS is that it requires specific knowledge, skills and experience for its use and data processing. Other limitations of the TLS are related to suitable scan positions for obtaining good visual angles of the glacier surface and stable scan positions for multi-temporal registration of repeated scans for change detection. In addition, the uncertainties of density conversion still remain at seasonal and annual scales as in situ measured densities of all benchmark sites are difficult to obtain (very sparse glaciers in China
have such detailed observations as Urumqi Glacier No.1). A day with little snow in the accumulation area and no snow in the

ablation area (i.e. snow line is clearly distinguished) should be chosen to perform TLS measurements. We may use a built-in camera of the TLS to create high resolution panorama images of a glacier (RIEGL Laser Measurement Systems, 2014a), then firn/snow and bare ice areas (i.e. snow line) can be determined (e.g. Barandun et al., 2018). Area-weighting approach can be used to estimate a density because the lack of in situ measured densities makes volume-weighting approach difficult to extensively use.

A density assumption over time intervals (≥5 years) based on physical models is also important as most glaciers in northwest China are cold and multi-thermal.

## 7 Conclusions

Urumqi Glacier No.1 is one of the reference glaciers in the WGMS network, a representative glacier in Central Asia and the best-monitored glacier in China. Here, for the first time, we have presented the potential of a novel long-range TLS to monitor summer

and annual geodetic mass balances of Urumqi Glacier No.1. The Riegl VZ®-6000 TLS has long scan range up to 6 km and is exceptionally well suited for measuring snowy and icy terrains in glacier mapping. We use TLS-derived DEMs to calculate summer and annual surface elevation changes and geodetic mass balances of the glacier for two consecutive years (2015-17) as well as to delineate accurate glacier boundaries.

Our analysis suggests that Urumqi Glacier No.1 has experienced pronounced thinning and mass loss for the four investigated periods. Glacier surface elevation lowering and mass loss during the summer were slightly greater than annual values. Glacier-wide geodetic mass balance in the mass-balance year 2015-16 was -0.72 ± 0.19 m w.e., which was slightly more negative than in the second mass-balance year. The majority of TLS-derived geodetic elevation changes at individual stakes were slightly positive, but insignificant compared to the glaciological elevation change (changes in exposed stake height) of individual stakes ($R^2 \geq 0.90$). The

difference in glacier-wide mass balances of Urumqi Glacier No.1 between the two methods was close to zero in 2016-17 but relatively larger differences were detected in summer 2015 for the whole glacier and east branch, which were related to the presence of fresh snow at the time of TLS surveys. Statistical analysis shows that agreement between the glaciological and geodetic methods can be considered as satisfying. Pronounced differences in spatial distributed mass balance mainly occurred at the steep elevations where in situ measurements were missing, which potentially induce the deviations in mass balance elevation distribution.

Despite uncertainties inherent in TLS-derived geodetic mass balances, our results show that the TLS device yields reliable results and is therefore well suited to the study of Urumqi Glacier No.1 since the observed vertical velocity component is small. Further more, the TLS can provide accurate and detailed information on glacier area and mass balance changes, its temporal-spatial resolution allows more detailed insight into the glacier's evolution. The greatest strength of the TLS is the long-range scanning which allows most of the glacier surface to be measured, including areas that are inaccessible for in situ measurements. Use of the

TLS-based geodetic method will be an important development since it is clearly a beneficial complement to direct glaciological mass balance, particularly for calibrating the unmeasured areas and validating the distributed glacier mass-balance models. A combination of glaciological and TLS observations may yield the optimum results. What's more, the TLS has application potential for glacier mass-balance monitoring in western China as most glaciers (~83% of the total number) have an area smaller than1 km$^2$. For a wide application of the long-range TLS, we can select some benchmark glaciers with easily accessible locations for TLS

measurements, but the presence of data voids and snow is still an enormous challenge, the quality of point cloud and DEM differencing and density conversion over short time intervals should be considered.

*Data availability.* Glaciological mass balance data related to this study are submitted to the WGMS and will be available at website: http://wgms.ch/. TLS point cloud data are available upon request by email to the corresponding author (lizq@lzb.ac.cn).

*Competing interests.* The authors declare that they have no conflict of interest.

*Acknowledgements.* This work is supported by the Strategic Priority Research Program of Chinese Academy of Sciences
(XDA20020102, XDA20060201), the Second Tibetan Plateau Scientific Expedition and Research (STEP) program (2019QZKK0201), National Natural Science Foundation of China (International cooperation and exchange projects) (41761134093), State Key Laboratory of Cryospheric Science (SKLCS-ZZ-2019) and National Natural Science Foundation of China (41471058, 41771081). Huilin Li also acknowledges the funding received from Key Research Program of Frontier Sciences of Chinese Academy of Sciences (QYZDB-SSW-SYS024). The authors are very grateful to Tien Shan Glaciological Station for the
continuous field observations. Additionally, the authors would like to thank the editor Tobias Bolch and referees for their numerous invaluable comments in improving the manuscript.

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

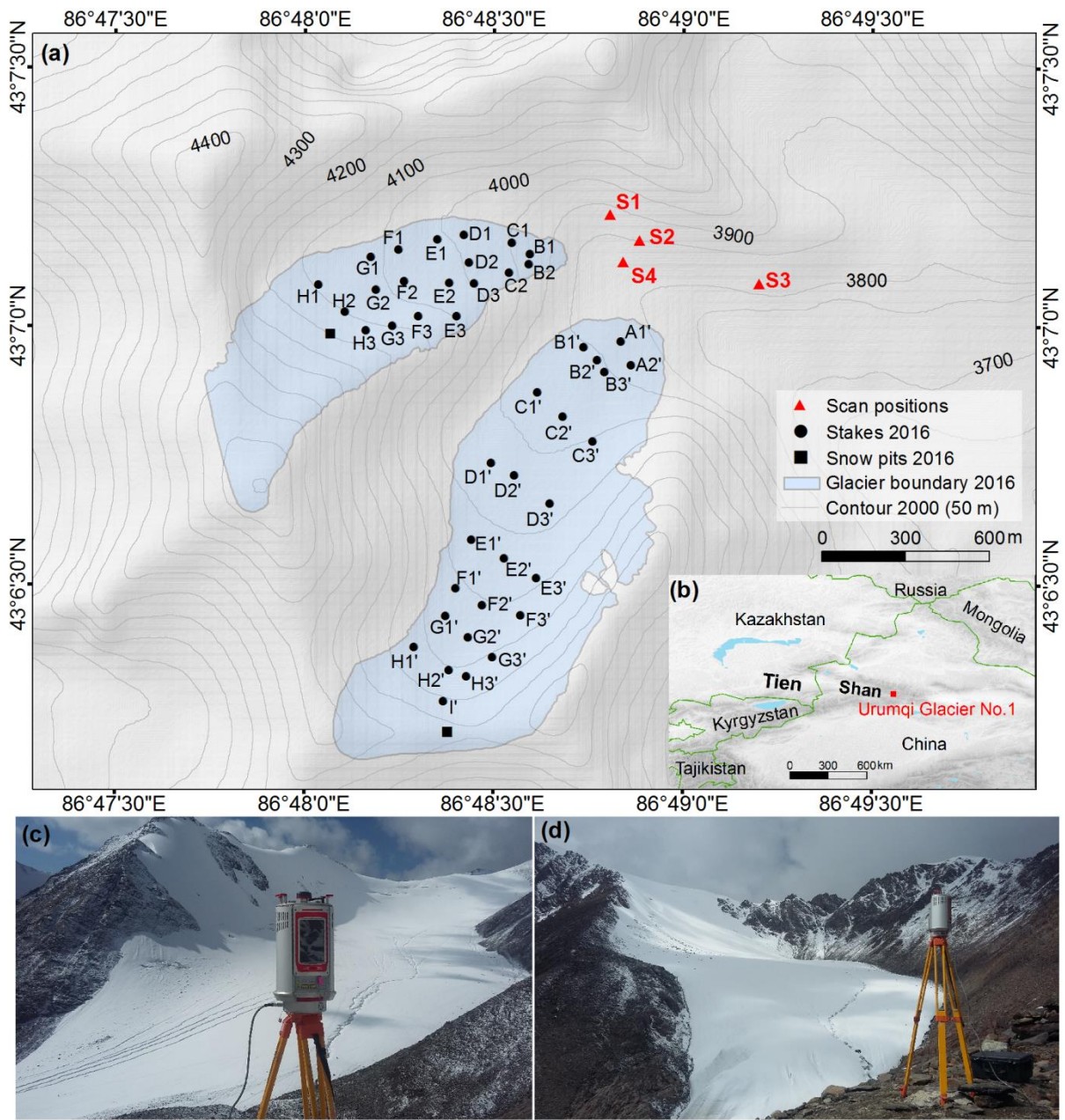

**Figure 1.** Overview of the study area. (a) The glaciological mass balance measuring network in 2016; glacier boundary delineated from TLS-derived DEM (1 September 2016) and spatial distributions of four scan positions. (b) Location map of Urumqi Glacier No.1 in eastern Tien Shan. (c) Riegl VZ®-6000 TLS survey of east branch at scan position S2 and (d) TLS survey of west branch at scan position S1 (27 August 2017).

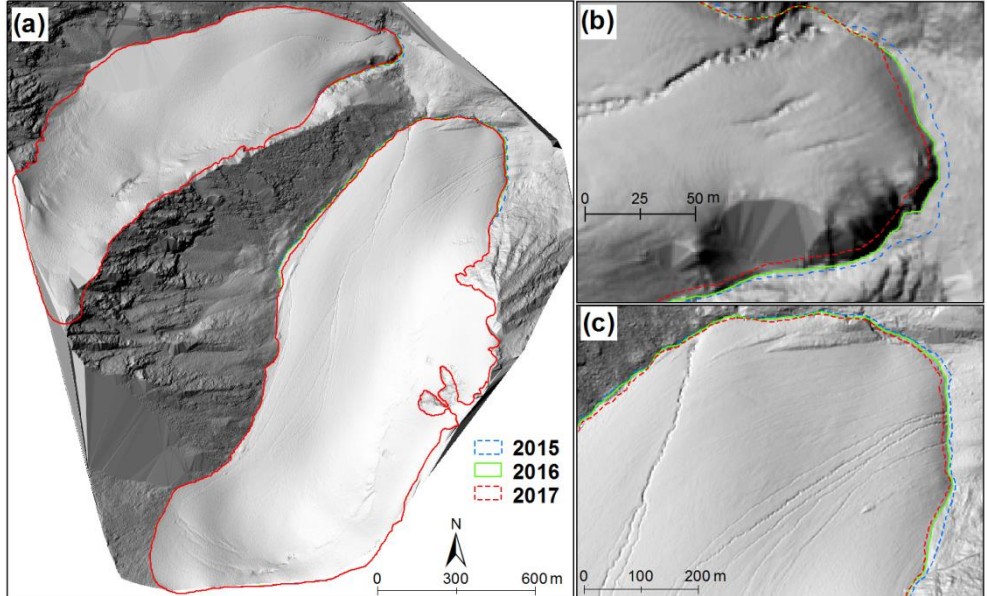

**Figure 2.** (a) Shaded reliefs of Urumqi Glacier No.1 margin calculated based on the TLS-derived DEM (on 1 September 2016) with the glacier boundary 2015 (blue), 2016 (green) and 2017 (red). Glacier terminus variations of west branch (b) and east branch (c) are also shown.

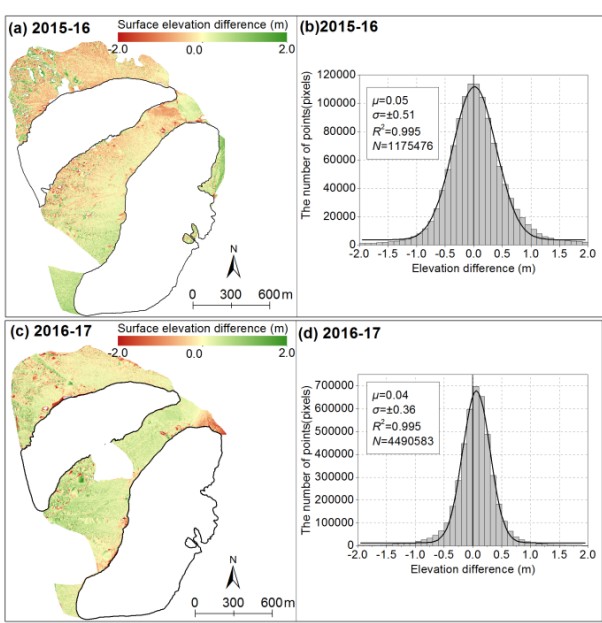

**Figure 3.** Statistics on annual surface elevation changes over stable terrain extracted by differencing of TLS-derived DEMs from two consecutive years. Spatial and corresponding frequency distributions of these changes for mass balance year 2015-16 (a, b), and 2016-17 (c, d). The median ($\mu$) and the standard deviation ($\sigma$) of the elevation differences, as well as the number of pixels ($N$) off glacier are given.

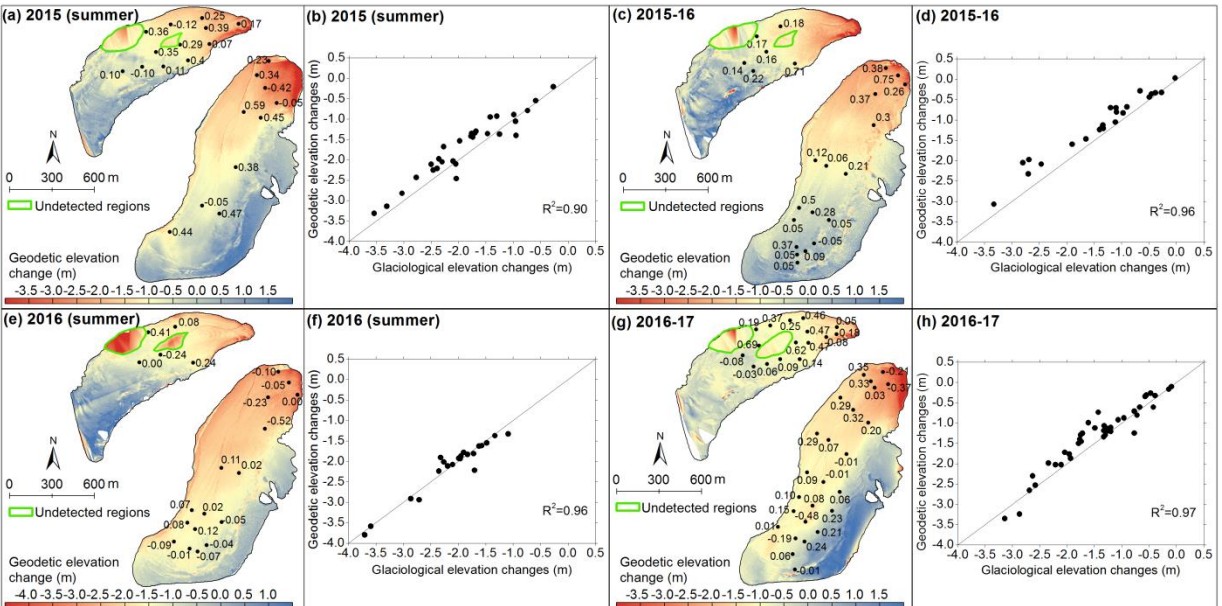

**Figure 4.** Spatial distribution of TLS-derived glacier surface elevation changes (a, c, e and g); the numbers represent the differences (in m) between the TLS-derived ($\Delta h_{\mathrm{TLS}}$) and glaciological in situ measured ($\Delta h_{\mathrm{glac}}$) elevation changes at corresponding ablation stakes ($\Delta h_{\mathrm{TLS}} - \Delta h_{\mathrm{glac}}$). Scatter plots of glaciological elevation change against geodetic elevation change at corresponding ablation stakes are presented, and the quality of fittings in terms of $R^2$ is also presented (b, d, f and h). Black lines are TLS-derived glacier boundary of Urumqi Glacier No.1, the boundaries in (a) and (c) are same as boundary 2015 in Fig. 2a, boundaries in (e) and (g) are from boundary 2016 and 2017 in Fig. 2a, respectively. White areas indicate outliers, which we have deleted. Two green polygons indicate areas that have not been detected by the TLS.

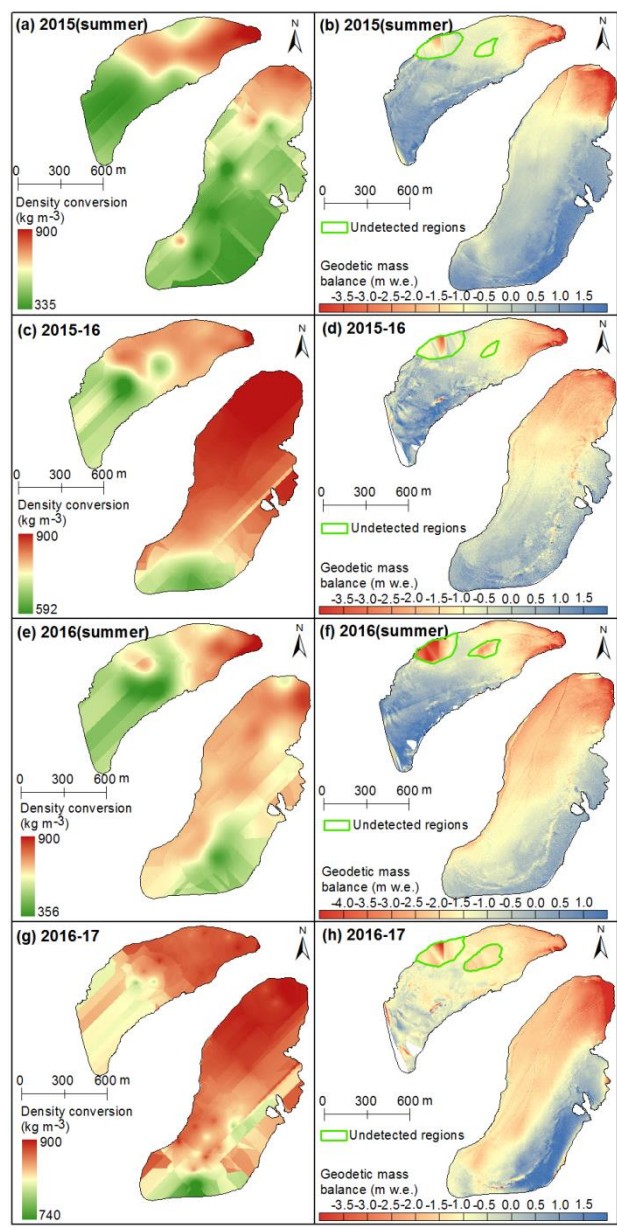

**Figure 5.** Distributed density conversions (a, c, e and g) and corresponding glacier-wide geodetic mass balance (b, d, f and h). Two green polygons indicate areas that have not been detected by the TLS.

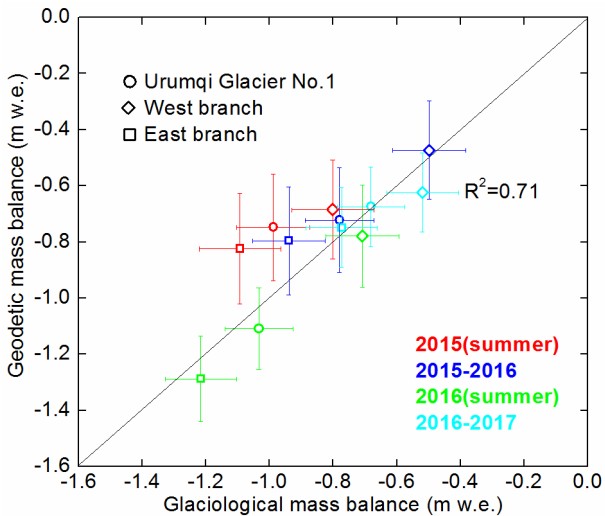

**Figure 6.** Glaciological versus TLS-derived geodetic mass balances for Urumqi Glacier No.1, the west branch and the east branch, with errors bars for two independent methods.

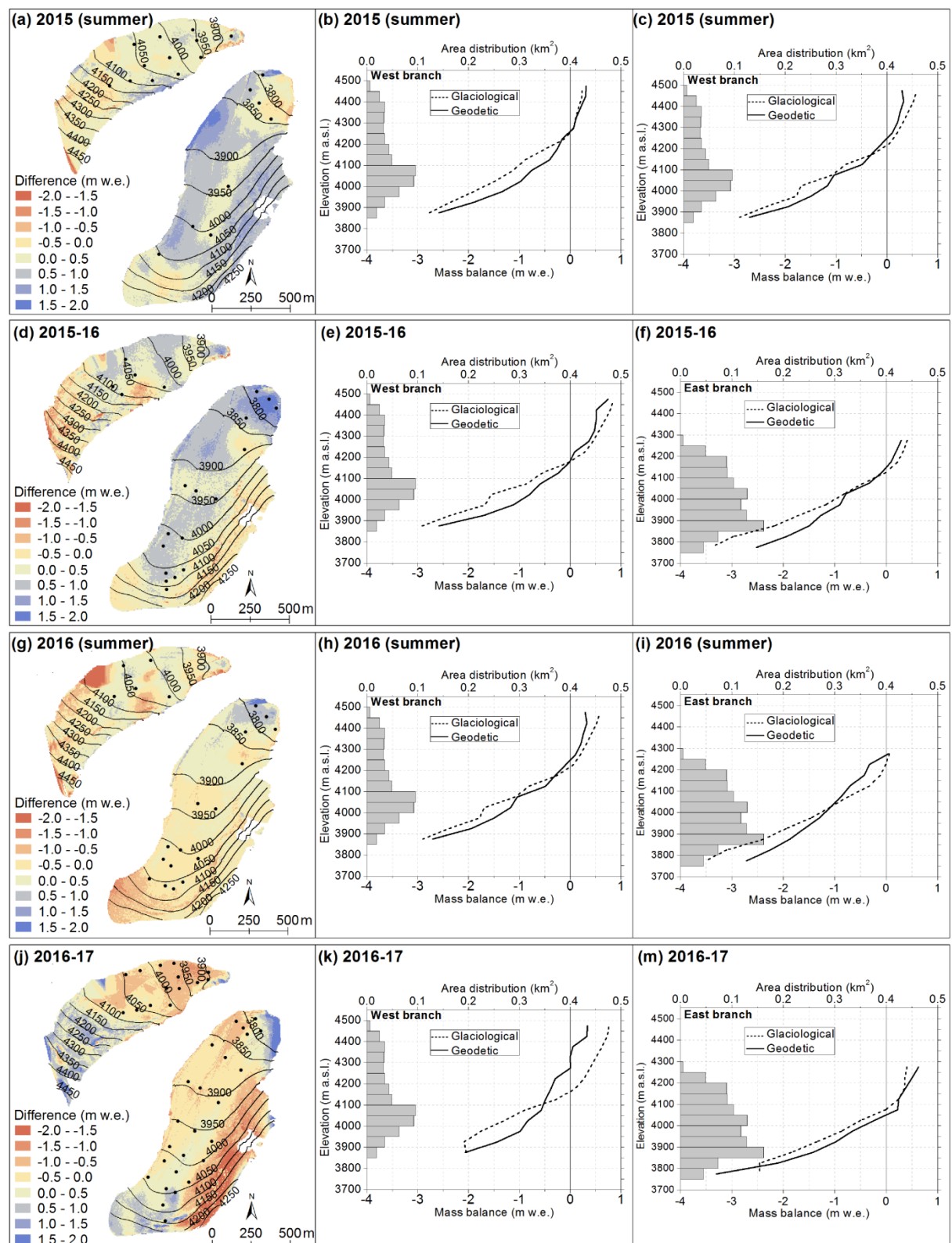

**Figure 7.** Spatial distributed difference derived from TLS-derived geodetic mass balance minus glaciological mass balance (a, d, g and j), black dots represent the location of well-measured ablation stakes, which are same as Fig. 5. The hypsometry (50 m altitudinal ranges) and the glaciological (dotted line) and geodetic (solid line) mass balance elevation distribution for the whole study period; both summer and annual mass balances are shown. Gray horizontal bars indicate the area-elevation distribution of Urumqi Glacier No.1. Note that the spatial resolution of glacier-wide geodetic mass balance in summer 2015 and 2016 was down-scaled to 5 m, in mass-balance years 2015-16 and 2016-17 was down-scaled to 7 m to coincide with glaciological glacier-wide mass balance.

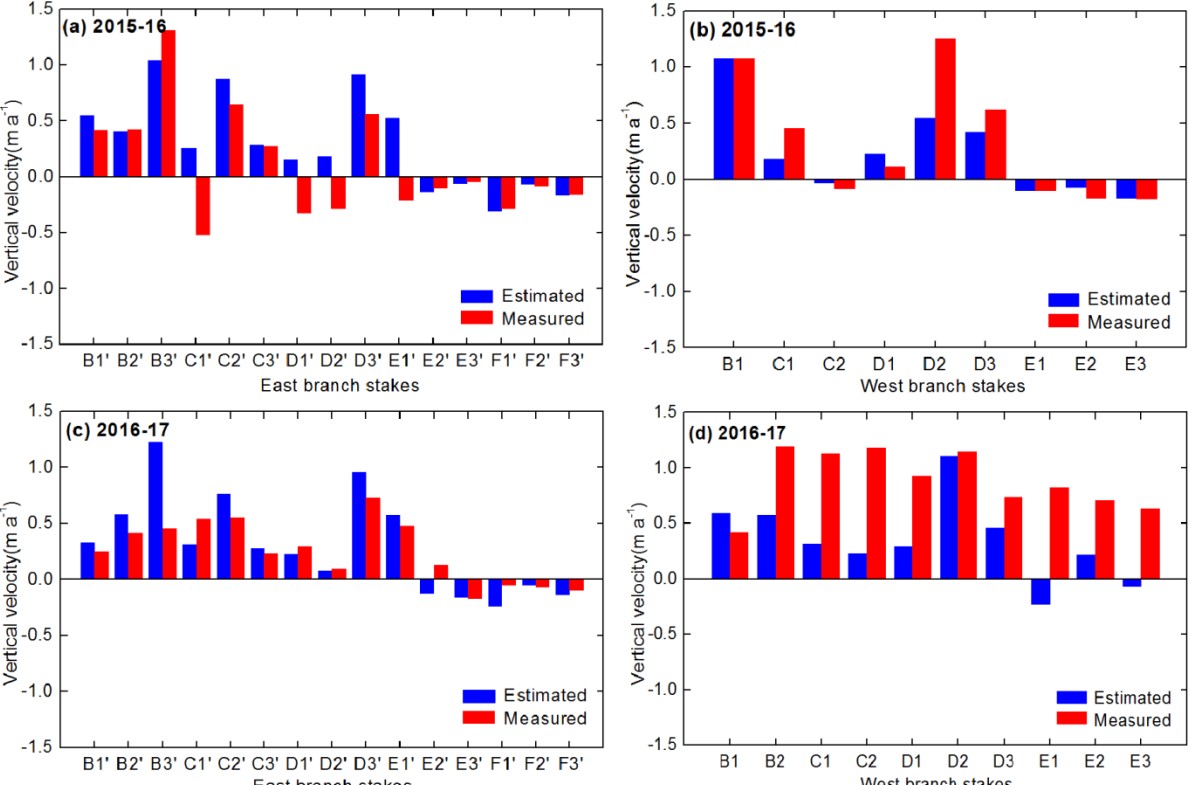

**Figure 8**. Comparison between estimated and in situ measured vertical velocity for the mass balance year 2015-16 and 2016-17; the letters represent ablation stakes (Fig. 1). Note than the summer periods and stakes in the higher elevations were not selected for comparisons due to snow cover reduced the quality of in situ measured vertical velocity.

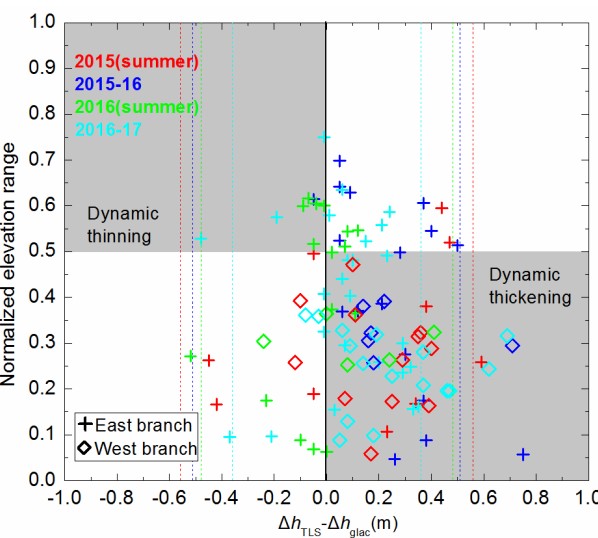

**Figure 9.** Changing differences between TLS-derived ($\Delta h_{\text{TLS}}$) and glaciological ($\Delta h_{\text{glac}}$) annual and summer surface elevation

10 changes at individual stakes versus normalized glacier elevation range of east branch and west branch for the four investigated

periods. Note than dash vertical lines indicate the uncertainty ranges ($\sqrt{\sigma^2_{\Delta h\text{TLS}} + (\sigma_a^{\text{ice}})^2 + (\sigma_a^{\text{firn}})^2}$) of the changing differences

($\Delta h_{\text{TLS}} - \Delta h_{\text{glac}}$), and grey quadrants indicate theoretical areas for Urumqi Glacier No.1 in equilibrium.

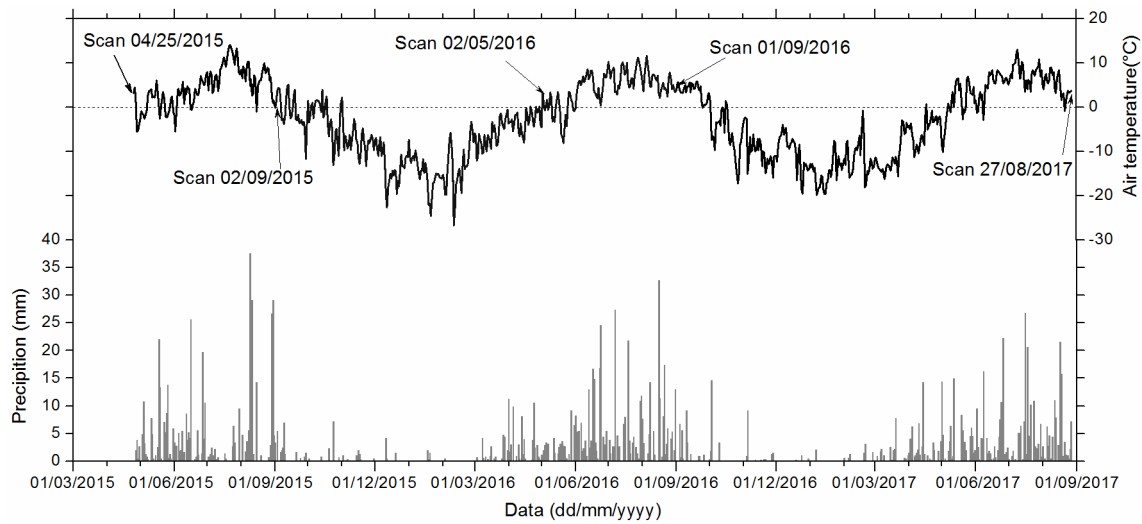

**Figure 10.** Daily precipitation and mean temperature observed at Daxigou Meteorological Station during 25 April 2015 - 28 August 2017.

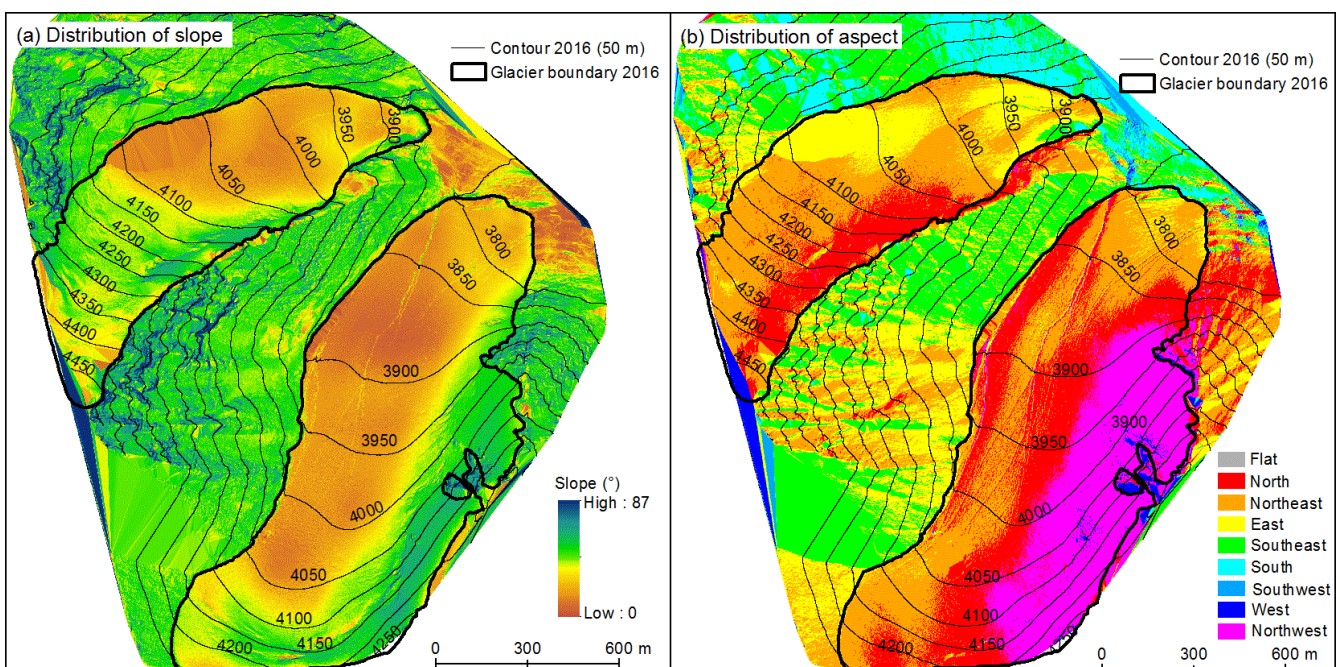

**Figure 11.** Spatial distributed slope (a) and aspect (b) of Urumqi Glacier No.1 extracted from TLS-derived DEM on 1 September 2016.

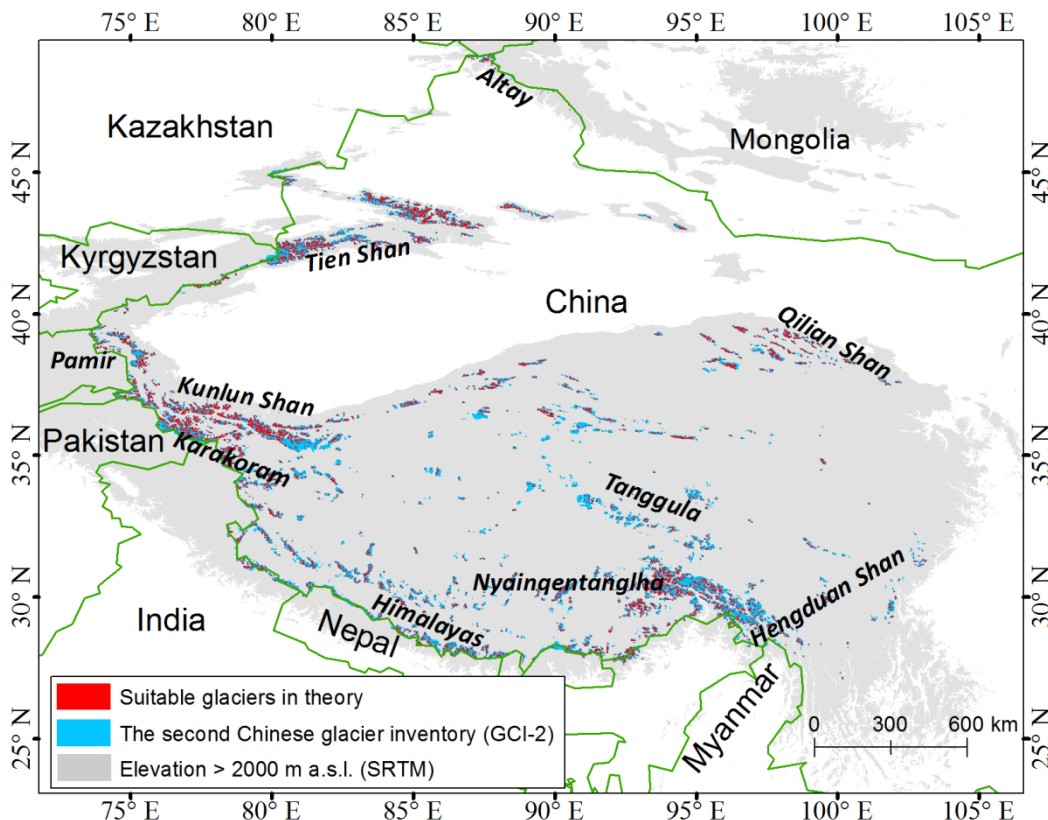

**Figure 12.** Spatial distribution of suitable glaciers in theory, those glaciers with an area of ≤ 1.6 km$^2$ (approximate area of Urumqi Glacier No.1) and a surface slope greater than 23.4 ° (mean slope of Urumqi Glacier No.1) have huge potential to be monitored using the TLS.

**Table 1.** Riegl VZ®-6000 TLS surveying parameters of Urumqi Glacier No.1.

| Date (dd/mm/yyyy) | Scanning range* (with overlap) (m²) | Number of points | Average point density (points m⁻²) | Vertical angle resolution (°) | horizontal angle resolution (°) | Total scan time (min) |
|---|---|---|---|---|---|---|
| 25/04/2015 | 3 204 684 | 12 740 500 | 3.98 | 0.020 | 0.020 | 46 |
| 02/09/2015 | 4 707 863 | 65 500 749 | 13.91 | 0.019/0.046 | 0.019/0.046 | 103 |
| 02/05/2016 | 3 224 285 | 26 908 210 | 8.35 | 0.020 | 0.020 | 82 |
| 01/09/2016 | 3 316 262 | 42 354 299 | 12.77 | 0.020 | 0.020 | 101 |
| 27/08/2017 | 3 161 489 | 54 835 821 | 17.34 | 0.020 | 0.020 | 88 |

*Scanning range is the total areas of four scan positions and does not include overlapped areas. The overlap percentage of the four scans on 25 April 2015 is smaller than other scan campaigns so that the average point density is relatively low.

**Table 2.** Error or StdDev ($\sigma_{MSA}$) of Multi-Station Adjustment (MSA) and the number of points ($n$) used for multi-temporal registration of two consecutive campaigns, the mean ($\mu$) and the standard error ($\sigma_{\overline{\Delta hTLS}}$) are measures of error derived by calculating elevation changes from TLS over stable terrain (off-glacier) for 2015 summer (25 April–2 September 2015), 2015-16 (2 September 2015–1 September 2016), 2016 summer (2 May–1 September 2016) and 2016-17 (1 September 2016–27 August 2017)

| Period | Error or Stdev of MSA (m) | Number of points | Mean elevation changes over stable terrain (m) | Standard error of elevation changes over stable terrain (m) |
|---|---|---|---|---|
| 2015 (summer) | 0.28 | 11 214 842 | -0.01 | 0.25 |
| 2015-16 | 0.07 | 10 182 829 | 0.05 | 0.23 |
| 2016 (summer) | 0.20 | 10 486 985 | -0.01 | 0.22 |
| 2016-17 | 0.07 | 18 657 232 | 0.04 | 0.16 |

**Table 3.** Glacier-wide mean of density conversion (ρ) and its uncertainty ($\sigma_\rho$) (in kg m$^{-3}$) as well as TLS-derived glacier surface elevation changes ($\overline{\Delta h \text{TLS}}$) (in m). TLS-derived geodetic ($B_{\text{geod}}$) and in situ measured glaciological ($B_{\text{glac}}$) net mass balance at winter and annual scales are listed (in m w.e.).

| Period | $\rho$ | $\sigma_\rho$ | $\overline{\Delta h \text{TLS}}$ | $B_{\text{geod}}$ | $B_{\text{glac}}$ |
|---|---|---|---|---|---|
| 2015 (summer) | | | | | |
| Urumqi Glacier No.1 | 752 | 34 | -0.991 | **-0.75 ±0.19** | **-0.99 ±0.12** |
| West branch | 696 | 35 | -1.014 | -0.68 ±0.18 | -0.80 ±0.13 |
| East branch | 782 | 33 | -0.952 | **-0.82 ±0.20** | **-1.09 ±0.13** |
| 2015-16 | | | | | |
| Urumqi Glacier No.1 | 810 | 21 | -0.827 | -0.72 ±0.19 | -0.78 ±0.11 |
| West branch | 763 | 24 | -0.625 | -0.47 ±0.18 | -0.50 ±0.12 |
| East branch | 837 | 20 | -0.873 | **-0.80 ±0.19** | **-0.94 ±0.11** |
| 2016 (summer) | | | | | |
| Urumqi Glacier No.1 | 622 | 32 | -1.654 | -1.11 ±0.15 | -1.03 ±0.11 |
| West branch | 579 | 34 | -1.230 | -0.78 ±0.13 | -0.71 ±0.12 |
| East branch | 647 | 31 | -1.925 | -1.29 ±0.15 | -1.22 ±0.11 |
| 2016-17 | | | | | |
| Urumqi Glacier No.1 | 864 | 19 | -0.746 | -0.68 ±0.14 | -0. 68 ±0.11 |
| West branch | 861 | 19 | -0.844 | -0.75 ±0.14 | -0.77 ±0.11 |
| East branch | 865 | 19 | -0.729 | -0.62 ±0.14 | -0.52 ±0.11 |