# Peer review of "Long-range terrestrial laser scanning measurements of annual and intra-annual mass balances for Urumqi Glacier No.1, eastern Tien Shan, China"

_The Cryosphere, 2018_

## Referee Comment (RC1) · Anonymous Referee #1 · 18 Sep 2018

Chunhai Xu and colleagues present a detailed reanalysis of annual and seasonal glaciological and geodetic balances at Urumqi Glacier No. 1, eastern Tien Shan, China, obtained between 2015 and 2017. This study puts a terrestrial laser scanner (TLS) dataset with high spatial and temporal resolution over the period of record at its value. The comparisons of geodetic results with the glaciological balances from an in-situ network are carried out in a thorough way and include an error assessment according to international practises.

The authors mentioned two publications using similar methods (Xu, C., Li, Z., Wang, F., Li, H., Wang, W., & Wang, L. (2017), doi:10.1017/jog.2017.45 and Xu, C., Li, Z., Wang,

P., Anjum, MN., Li, H., & Wang, F. (2018), doi:10.1016/j.coldregions.2018.08.006), which can be seen as preliminary studies to the proposed manuscript. Hence, the discussion paper has been cross-read with the mentioned publications in terms of basic quality issues concerning significance, originality and novelty of the study.

Reading the papers, I had the impression that many sections are redundant. Besides the Introduction and Study site sections, the "Data and Methods" and "Uncertainty" chapters also seem to be similar, show no new insights and could at least be omitted by referencing. Furthermore, the Conclusions have redundant elements to the other two studies. Working through the manuscript new information is only provided by i) altering the temporal scale, ii) introducing an approach of density conversion and iii) the consideration of internal processes when comparing the two methods. Although the authors state that they implement a detailed comparison between glaciological and geodetic mass balances at seasonal and annual scales and assess the potential of a novel long-range TLS to monitor glacier mass balance, the obvious redundancy puts the manuscript on the fringe of acceptance.

Weighing up these points, I think that the new information provided in this Discussion paper is not sufficient or suitable for publication.

---

## Referee Comment (RC2) · Anonymous Referee #2 · 4 Nov 2018

General comments: This study describes the use of a novel long-range terrestrial laser scanner (TLS) dataset to calculate annual and summer mass balances and delineate accurate glacier boundary of Urumqi Glacier No.1 in eastern Tien Shan over two consecutive years (2015-17). After introduction of the data used and methodology applied, the authors showed TLS-derived surface elevation and geodetic mass changes. They then compared these results with the conventional glaciological method following the framework proposed by Zemp et al (2013) to validate the accuracy and relevance of the TLS to monitor glacier mass balance. At the end, they give a discussion about the quality of TLS data and DEM differencing, explain the possible causes about differences between the two methods and evaluate the potential of such long-range TLS

to measure seasonal and annual glacier mass balance. The paper employs advanced instrument of glacier mass balance monitoring and can be seen as a deep-going study of a published paper (Xu et al., 2017, J. Glaciol., doi:10.1017/jog.2017.45). China contains the largest number of glaciers outside the polar regions, very few glaciers have discontinuous glaciological mass balance records, so we need alternative approaches that could complement glaciological method. The presented study is very interesting, which I think to be a valuable contribution to The Cryosphere. However, there are some comments and issues that the authors should be addressed. 1) The discussion about the potential of the long-range TLS to measure glacier mass balance is very weak, which undermines the paper. Please see detailed comments in "Specific " 2) Uncertainty assessment of the glaciological methods: you have quantified various errors according to other similar studies, but I firmly believe that these values are really different, especially for errors in spatial extrapolation over the entire glacier. Just as you say relative smaller area and accompanying higher density of point measurements of UG1 than other glaciers decide the uncertainty is smaller. Could you compare specific net mass balance with in situ measured stake datasets of UG1 to determine the error of spatial interpolation? 3) Some sentences should be written more clearly and precisely, including P2, L4; P2, L34; - P6, L26; . . . 4) Figures: Figures need some improvements in terms of visibility of their content.

Specific comments: 0 Abstract - P1, L10: Delete "typically". To date the glaciological method is commonly used to measure seasonal and annual surface mass balance. So it is not necessary to emphasize the method using "typically". - P1, L10: Rephrase "seasonal surface mass balance" - P1, L11: Replace "measuring networks" with "field networks" - P1, L15: "scanner" instead of "scanning"

1 Introduction - P2, L4: Add "are spare and discontinuous". Please rephrase to be more accurate. - P2, L7: I would rather delete "entire". It is not always possible to cover the entire glacier, such as ICESat. - P2, L12: Replace "spatiotemporal" by "time". I know some images have high spatial resolution at present, e.g. Cartosat-1

(2.5 m), Pléiades (0.5 m), QuickBird(0.61 m), GeoEye(0.41-1.65 m) etc. - P2, L28-29: Rephrase "...central and bottom elevations were detected due to the glacier area is relatively small." - P2, L34: What the meaning of "best-monitored glacier"? I guess you mean Urumqi Glacier No.1 has the longest and most detailed surface mass balance measurements in China. Please rephrase the sentence to be clear. - P2, L38-40: Rephrase "To date, comparison of glaciological and geodetic mass balances ...for the period 1981-2009 at intervals of several years, geodetic reanalysis of seasonal and annual glaciological mass balance..." - P2, L40: You already have a publication about the reanalysis of glaciological and geodetic mass balances of UG1 (Xu et al., 2018, Cold Reg. Sci. Technol., doi: 10.1016/j.coldregions.2018.08.006), please write here.

2 Study site - P3, L23: Add "...a northeast-orientated small..." - P3, L24: Replace "Fig" by "Figs" - P3, L24 "and consists of two independent small glaciers: the east branch (EB) and the west branch (WB)" would be better at the end of this paragraph and then delete "and consists of two independent small glaciers" - P3, L27 "long-term measurements"? I think you may mean something like glacier mass balance? - P3, L27 Correct "Over the past 50 years" and give a specific time period. - P3, L31-37 I suggest that related literatures should be cited here. - P4, L4-7 This paragraph would be better in section 3.3 as it already mentions methodology

3 Data and methodology - P4, L25: Replace "Fig" by "Figs" - P4, L26: Everywhere else in the manuscript, please replace "GPS" by "GNSS" - P5, L1: Add "...in the range of..." - P6, L9: It seems the authors mixed the triangle (âŰş) and capital Greek letter delta ($\triangle$), and whole manuscript: please replace "âŰş" by "$\triangle$". - P6, L26: Please rephrase "volume changes are considerable" to be more precise. - P6, L36: Figs - P7, L7: Add "...the glacier and evenly distributed..." - P7, L18: I know what you meaning of "the specific mass balance is calculated from the product of the level change between readings and the ice density" as I have calculated glaciological surface mass balance, but it is not easy to understand for wide readers. Please rewrite the sentence to be clearer. - P7, L29: Which energy-balance model? Please give a brief introduction and

refer corresponding literatures. - P8, L3: Correct "Figs"

4 Uncertainty assessments - P8, L19: Add "windless weather conditions" - P8, L31: Can you give absolute values of the proportions of the two artefacts over the entire glacier and then quantify the errors related to unscanned areas. - P9, L13: The given errors can be listed with 2 decimal places to reflect appropriate level of certainty. - P9, L35: You should write clearly here that the value you cited indicates point mass balance. - P10, L4: What the meaning of sampling sites? Please rephrase to be more specific - P10, L4: You mean something was given in Table 4? Add some sentences to be clear.

5 Results - P10, L23: Replace "clear" by "clearer" - P10, L29: Correct "Figs" - P10, L29-30: I know debris cover on a glacier may alleviate ablation when the debris thickness exceeds a certainty value. But your argument explaining the phenomena is quite speculative. Please support your opinion by some semi-quantitative or quantitative data. - P10, L33: Use "with" instead of "by" - P10, L33: Please correct "Figs" - P11, L2: Again: please correct "Figs" - P11, L7: Rephrase "...all of the four investigated periods" - P11, L9: It makes no sense that the value is rounded to three decimal places, please change everywhere else in the manuscript. - P11, L11: Add "compared to the corresponding values of EB" after "... more negative" - P11, L16: Fig. 5 instead of Fig 7? - P11, L17: Replace "sites" with "ablation stakes"

6 Discussion - P12, L30: Add "of each scan positions" - P12, L30: Fig. 4 instead of Fig. 3? - P12, L32: Replace "are" with "is" - P13, L9: I know the number and location of ablation stakes vary from year to year as stakes melt out and sink. Please give a specific period for the average value. - P13, L11: Delete "in" - P13, L13; How did you decide the annual discharge? I guess you use the mean value here, can you calculate the internal and basal ablation using the measured data of each year? - P13, L32: I would delete the first sentence in this paragraph as it had appeared in the introduction. - P14, L7-9: Can you quantify the influences of unscanned areas? - P14, L10: Rephrase "a discrepancy in mass balance elevation distributions of WB was observed
at. . ." - P14, L11: Replace "takes" with "take" - P14, L33: The discussion about the potential of the long-range TLS to measure glacier mass balance is very weak and not really satisfying, but I believe that there is more to say. e.g. Comparison with other technologies, such as unmanned aerial systems, terrestrial photogrammetry, and then you can discuss the advantages/disadvantages of each technology. We see some data voids; can you say something about future application of such TLS to monitor glacier evolution. I firmly sure that artefacts will also exist for other glaciological applications. The data voids can be avoided when combining with other approaches? Density conversion is still a challenge at annual and seasonal scales, which assuredly influences the wide application. What do you advise as reduction of the density conversion? I don't really think the majority of glaciers can be measured using the TLS as some of them lie at remote locations. I would rather suggest you to select some representative glaciers (evenly distributed at different mountains, different types and areas, etc.) with easily accessible locations for the geodetic mass balance monitoring. Can you discuss something about application of TLS to monitor the representative glaciers; it would be very interesting and relevant to know additional information of those glaciers for future studies. Can you give more information about TLS-derived geodetic results to validate the distributed glacier mass-balance models; I think it is very important for future glaciological studies since its high spatiotemporal resolution and the shortage of in situ measurements. - P15, L9: I think microwave remote sensing is not an effective technology as the limited time and space resolution. 7 Conclusions Cloud need to be a bit changed after taking account the comments mentioned above. 8 Figures and tables - Figure 2: In the caption, please add some scientific content to illustrate the figure. - Figure 3: Please improve the figure to obtain clear content. - Figure 4: Please again improve the figure to obtain clear content. - Figure 5: Please again improve the figure to obtain clear content. - Figure 6: Please again improve the figure to obtain clear content. - Table 3: Please hold two decimal places. - Table 4: Please again hold two decimal places.

9 References Please check the reference, both in the text and at the end, to meet the

requirements of the journal. e.g.: - P1, L29: Correct "Liu and Liu, 2016" - P2, L2: Correct "Xie and Liu, 1991" - P18, L19: Lichti et al., 2005 in references not in text - P19, L12: Rolstad after RIEGL - etc.

---

## Author Comment (AC1) · 3 Dec 2018

We'd like to thank the referee for the valuable, constructive and detailed comments which certainly helped to improve the manuscript. The corresponding changes and refinements have been made in the revised paper (track changes was used in order to be easily identified) and are also summarized in our reply below. Reviewer comments in normal font, our reply to each comment is provided after the comment and given in bold font.

Reply to comments from anonymous referee 1

[Figure]

Chunhai Xu and colleagues present a detailed reanalysis of annual and seasonal glaciological and geodetic balances at Urumqi Glacier No. 1, eastern Tien Shan, China, obtained between 2015 and 2017. This study puts a terrestrial laser scanner (TLS) dataset with high spatial and temporal resolution over the period of record at its value. The comparisons of geodetic results with the glaciological balances from an in-situ network are carried out in a thorough way and include an error assessment according to international practises. The authors mentioned two publications using similar methods (Xu, C., Li, Z., Wang, F., Li, H., Wang, W., Wang, L. (2017), doi:10.1017/jog.2017.45 and Xu, C., Li, Z., Wang, P., Anjum, MN., Li, H., Wang, F. (2018), doi:10.1016/j.coldregions.2018.08.006), which can be seen as preliminary studies to the proposed manuscript. Hence, the discussion paper has been cross-read with the mentioned publications in terms of basic quality issues concerning significance, originality and novelty of the study. Reading the papers, I had the impression that many sections are redundant. Besides the Introduction and Study site sections, the "Data and Methods" and "Uncertainty" chapters also seem to be similar, show no new insights and could at least be omitted by referencing. Furthermore, the Conclusions have redundant elements to the other two studies. Working through the manuscript new information is only provided by i) altering the temporal scale, ii) introducing an approach of density conversion and iii) the consideration of internal processes when comparing the two methods. Although the authors state that they implement a detailed comparison between glaciological and geodetic mass balances at seasonal and annual scales and assess the potential of a novel long-range TLS to monitor glacier mass balance, the obvious redundancy puts the manuscript on the fringe of acceptance. Weighing up these points, I think that the new information provided in this Discussion paper is not sufficient or suitable for publication.

Reply: Thanks for the careful reading! As a matter of fact, the scientific achievements of three publications are totally different. The achievements of the first published paper (2017 in Journal of Glaciology) are to evaluate accuracy and precision of glacier surface elevation changes retrieved from long-range terrestrial laser scanner (TLS), and

to test applicability of such TLS to monitor the mass balance of Urumqi Glacier No.1. Whether agreement between the glaciological and TLS-derived glacier-wide mass balance was pending, potential of such technology applied in seasonal and annual glacier mass-balance measurements in western China had not been assessed. The second publication (2018 in Cold Regions Science and Technology) presents a comparison between cumulative direct glaciological and geodetic mass balance data from 1981 to 2015 for Urumqi Glacier No. 1, so the achievements of the paper are to reanalyze the glaciological mass balance series. In order to achieve the achievements, we try to define the source of the observed uncertainties in glaciological and geodetic methods, and the extent of the mass balances influenced by the different DEMs used, the existing snow cover, the reference area and processes of internal accumulation and ablation.

At present, comparison of glaciological and geodetic mass balances mainly focuses on sub-decadal to decadal scales as the available DEMs usually limit the spatiotemporal resolution of geodetic mass-balance measurements, while seasonal and annual scales have received little attention. This paper uses a long-range TLS to monitor the summer and annual mass balance of Urumqi Glacier No.1 (UG1) as well as delineating accurate glacier boundaries for two consecutive years (2015-17), and discusses the potential of such technology in glaciological applications. Hence, the scientific achievements (aims) of the present study are: (1) to describe the original use of Riegl VZ$^{®}$-6000 TLS-derived DEMs to calculate summer and annual geodetic mass balances of UG1 for two consecutive years (2015-17); (2) to consider three-dimensional (3-D) changes of ice and firn/snow bodies and density conversion from in situ measured snow/firn densities is applied to make these calculations. Firn compaction and metamorphosis can be therefore captured to some extent; (3) to compare the geodetic results to glaciological glacier-wide mass balances through a detailed uncertainty assessment of the glaciological and geodetic methods; (4) to discuss how to achieve good quality of point cloud data and DEM differencing and to analyze the possible cause of the difference

between the two methods; and (5) to take UG1 as a case to assess the potential of such long-range TLS to measure glacier mass balance at the seasonal and annual scales and put forward some main considerations for a broader application of the TLS.

I agree that some chapters of the three publications seem to be similar, especially for "Study site" and "Data and methodology". However, the introduction of the presented study is substantially different from the two others as the different aims of each paper. In the section of Data and methodology, the descriptions of TLS and its data processing (subsection 3.1) as well as Uncertainty assessments (section 4) are more detailed and perfect than past studies, so we have not simply cited the two published papers. The methods of glaciological and geodetic mass balance calculations have been widely used for many publications, especially for a conceptual framework proposed by Zemp et al (2013). Hence we also directly referred the conceptual framework although the contents seem to be similar to our previous papers. Besides introducing an approach of density conversion at seasonal and annual scales, the present study also describes the delineation of accurate glacier boundary of Urumqi Glacier No.1, which updates and corrects previous published boundary (e.g. Wang et al., 2016; Xu et al., 2017). In addition, we implement a detailed comparison between direct glaciological and TLS-derived geodetic mass balance, including glacier-wide mass balances and mass balance elevation distributions derived from the two methods. In section 6, the discussion of data quality and DEM differencing is more in-depth than the first published paper. In the revised manuscript, we added new information to discuss the potential of the long-range TLS, including: 1) advantages and disadvantages between the long-range TLS and other technologies; 2) how to deal with data voids in future application of such TLS; 3) how to reduce the uncertainty of seasonal and annual density; and 4) application of TLS-derived geodetic results to validate the distributed mass-balance model. We hope the revised manuscript is suitable for publication.

With best regards, Chunhai Xu et al.

---

## Author Comment (AC2) · 3 Dec 2018

The authors would like to sincerely thank the referee for the valuable, constructive and detailed comments which certainly helped to improve the manuscript. The corresponding changes and refinements have been made in the revised paper (track changes was used in order to be easily identified) and are also summarized in our reply below.

Reply to comments from anonymous referee #2

General comments:

This study describes the use of a novel long-range terrestrial laser scanner (TLS) dataset to calculate annual and summer mass balances and delineate accurate glacier boundary of Urumqi Glacier No.1 in eastern Tien Shan over two consecutive years (2015-17). After introduction of the data used and methodology applied, the authors showed TLS-derived surface elevation and geodetic mass changes. They then compared these results with the conventional glaciological method following the framework proposed by Zemp et al (2013) to validate the accuracy and relevance of the TLS to monitor glacier mass balance. At the end, they give a discussion about the quality of TLS data and DEM differencing, explain the possible causes about differences between the two methods and evaluate the potential of such long-range TLS to measure seasonal and annual glacier mass balance. The paper employs advanced instrument of glacier mass balance monitoring and can be seen as a deep-going study of a published paper (Xu et al., 2017, J. Glaciol., doi:10.1017/jog.2017.45). China contains the largest number of glaciers outside the polar regions, very few glaciers have discontinuous glaciological mass balance records, so we need alternative approaches that could complement glaciological method. The presented study is very interesting, which I think to be a valuable contribution to The Cryosphere. However, there are some comments and issues that the authors should be addressed.

1) The discussion about the potential of the long-range TLS to measure glacier mass balance is very weak, which undermines the paper. Please see detailed comments in "Specific"

Reply: The very weak discussion has been enriched as suggested; please see the revised version of our manuscript.

2) Uncertainty assessment of the glaciological methods: you have quantified various errors according to other similar studies, but I firmly believe that these values are really different, especially for errors in spatial extrapolation over the entire glacier. Just as you say relative smaller area and accompanying higher density of point measurements of UG1 than other glaciers decide the uncertainty is smaller. Could you compare specific net mass balance with in situ measured stake datasets of UG1 to determine the error of spatial interpolation?

Reply: Thanks for the good comments. We have compared glacier-wide mass balance with individual sites. We find that the differences between specific net mass balance at individual sites and in situ measured point mass balance at corresponding sites were in the range of 0-0.042 m w.e. with an average value of 0.01 m w.e., namely, the error of spatial interpolation in the measured area is small. Therefore the error mainly originates from unmeasured areas

(e.g. accumulation areas), however, the lack of measured data in the accumulation areas limits us to quantify the error. We conservatively cite an empirical value from similar literature.

Now the paragraph was revised as:

The class (ii) errors originate from extrapolating observed values to unmeasured areas, insufficient spatial distribution of measured sites and the interpolation method. Hock and Jensen (1999) evaluated the error of the interpolation method at about ±0.1 m w.e. a-1 for mean specific mass balances. Huss et al. (2009) computed and compared mean specific net balance with randomly reduced annual stake datasets and found that the error was ±0.12 m w.e. a-1. For UG1, we find that the differences between specific net mass balance at individual sites and in situ measured point mass balance at corresponding sites were in the range of 0-0.042 m w.e. with an average value of 0.01 m w.e., namely, the error of spatial interpolation in the measured area is small. the firn basin and glacier tongue terrain of the WB are very steep and the upper eastern elevation of the EB is also precipitous, resulting in no in situ measurements are available in theses inaccessible areas. Therefore the error mainly originates from unmeasured areas (e.g. accumulation areas), however, the lack of measured data in the accumulation areas limits us to quantify the error. We conservatively assume that the corresponding uncertainty σ_extra was ±0.1 m w.e. a-1 (cf. Andreassen et al., 2016).

3) Some sentences should be written more clearly and precisely, including P2, L4; P2, L34; - P6, L26
Reply: Relevant sentences have been rewritten as suggested.

4) Figures: Figures need some improvements in terms of visibility of their content.
Reply: Done.

Specific comments:
0 Abstract
- P1, L10: Delete "typically". To date the glaciological method is commonly used to measure seasonal and annual surface mass balance. So it is not necessary to emphasize the method using "typically".
Reply: We fully agree and delete accordingly.

 - P1, L10: Rephrase "seasonal surface mass balance"
Reply: Done.

- P1, L11: Replace "measuring networks" with "field networks"
Reply: Done.

- P1, L15: "scanner" instead of "scanning"
Reply: Done.

1 Introduction
- P2, L4: Add "are spare and discontinuous". Please rephrase to be more accurate.
Reply: Now added accordingly.

- P2, L7: I would rather delete "entire". It is not always possible to cover the entire glacier, such as ICESat.

Reply: We agree! Now deleted as suggested.

- P2, L12: Replace "spatiotemporal" by "time". I know some images have high spatial resolution at present, e.g. Cartosat-1 (2.5 m), Pléiades (0.5 m), QuickBird(0.61 m), GeoEye(0.41-1.65 m) etc.

Reply: We agree and rephrase as suggested.

- P2, L28-29: Rephrase "…central and bottom elevations were detected due to the glacier area is relatively small."

Reply: Here I think the glacier size is big, so the sentence was rewritten as "…only the central and bottom elevations were detected due to the glacier area is relatively big."

- P2, L34: What the meaning of "best-monitored glacier"? I guess you mean Urumqi Glacier No.1 has the longest and most detailed surface mass balance measurements in China. Please rephrase the sentence to be clear.

Reply: Now this sentence is changed accordingly. "Urumqi Glacier No.1 (hereafter known as UG1) has the most detailed annual and seasonal surface mass balance measurements in China."

- P2, L38-40: Rephrase "To date, comparison of glaciological and geodetic mass balances …for the period 1981-2009 at intervals of several years, geodetic reanalysis of seasonal and annual glaciological mass balance…"

Reply: Now replaced accordingly.

- P2, L40: You already have a publication about the reanalysis of glaciological and geodetic mass balances of UG1 (Xu et al., 2018, Cold Reg. Sci. Technol., doi: 10.1016/j.coldregions.2018.08.006), please write here.

Reply: We have written as suggested.

2 Study site
- P3, L23: Add "…a northeast-orientated small…"
Reply: Done.

- P3, L24: Replace "Fig" by "Figs"
Reply: Done.

- P3, L24 "and consists of two independent small glaciers: the east branch (EB) and the west branch (WB)" would be better at the end of this paragraph and then delete "and consists of two independent small glaciers"

Reply: We have changed accordingly.

- P3, L27 "long-term measurements"? I think you may mean something like glacier mass balance?

Reply: Yes! Now revised as "…long-term glaciological mass-balance measurements."

- P3, L27 Correct "Over the past 50 years" and give a specific time period.

Reply: We have referred the literature and give a specific time period from 1959-2008.

- P3, L31-37 I suggest that related literatures should be cited here.

Reply: Now cited related literatures as follows:

References:

Li, Z., Li, H., and Chen, Y.: Mechanisms and simulation of accelerated shrinkage of continental glaciers: a case study of Urumqi Glacier No. 1 in Eastern Tianshan, central Asia. J. Earth Sci., 22, 423–430. http://dx.doi.org/10.1007/s12583-011-0194-5, 2011.

Liu, C., and Han, T.: Relation between recent glacier variations and climate in the Tien Shan mountains, Central Asia. Ann. Glaciol. 16, 11–16, 1992.

Han, T., Ding, Y., Ye, B., Liu, S., and Jiao, K.: Mass-balance characteristics of Urumqi Glacier No. 1, Tien Shan. China. Ann. Glaciol. 43, 323–328, 2006.

Huintjes, E., Li, H., Sauter, T., Li, Z., and Schneider, C.: Degree-Day Modelling of the Surface Mass Balance of Urumqi Glacier No. 1, Tian Shan, China. The Cryosphere Discussions, 4, 207–232, 2010.

- P4, L4-7 This paragraph would be better in section 3.3 as it already mentions methodology
Reply: Now removed this paragraph and changed section 3.3 accordingly.

3 Data and methodology
- P4, L25: Replace "Fig" by "Figs"
Reply: Replaced.

- P4, L26: Everywhere else in the manuscript, please replace "GPS" by "GNSS"
Reply: Replaced.

- P5, L1: Add "…in the range of…"
Reply: Done.

- P6, L9: It seems the authors mixed the triangle (â″U¸s) and capital Greek letter delta (Δ), and whole manuscript: please replace "â″U ¸s" by "Δ".
Reply: Replaced accordingly.

- P6, L26: Please rephrase "volume changes are considerable" to be more precise.

Reply: Now revised as "volume changes significantly different from zero"

- P6, L36: Figs

Reply: Done.

- P7, L7: Add "…the glacier and evenly distributed…"

Reply: I think here is L12, now added.

- P7, L18: I know what you meaning of "the specific mass balance is calculated from the product of the level change between readings and the ice density" as I have calculated glaciological surface mass balance, but it is not easy to understand for wide readers. Please rewrite the sentence to be clearer.

Reply: Subsection 3.3.1 is about glaciological measurements, so we have removed the sentence into subsection 3.3.1, and then the sentence was revised as:

Glaciological mass balance includes point and glacier-wide mass balances. The rate of mass gain and loss per unit time is accumulation rate $\dot{c}$ and ablation rate $\dot{a}$, respectively, $\dot{c}$ minus $\dot{a}$ equals mass-balance rate $\dot{b}$. Integrating $\dot{b}$ over the time span from $t_0$ to $t_1$ gives point mass balance $\Delta b$

- P7, L29: Which energy-balance model? Please give a brief introduction and refer corresponding literatures.

Reply: We revised the sentence as "together with simulated values obtained using a simple energy-balance model (the energy divide into shortwave radiation and temperature dependent energy budget) in areas with no measurements (Oerlemans, 2010; WGMS, 2017)"

In the revised manuscript we refer corresponding literature as follows:

Reference:

Oerlemans., J.: The Microclimate of Valley Glaciers, Igitur, Utrecht Publishing and Archiving Services, Universiteitsbibliotheek Utrecht, Utrecht, 2010.

- P8, L3: Correct "Figs"

Reply: Corrected.

4 Uncertainty assessments

- P8, L19: Add "windless weather conditions"

Reply: Added.

- P8, L31: Can you give absolute values of the proportions of the two artefacts over the entire glacier and then quantify the errors related to unscanned areas.

Reply: Good comments! We had delineated unscanned regions and the corresponding proportions of the two artefacts over the glacier surface were in the range of 3.1-4.6%. The

lack of dense measured 3-D coordinates of the two artefacts limits us to assess terrain-induced errors quantitatively. The artefacts were not taken into account in calculating the mass balance in order to be precise, but the errors related to unscanned areas should be very small because the proportions of the two artefacts were minor. This did not influence a direct comparison between glaciological and geodetic mass balances.

Now the paragraph was revised as:

For precision, the artefacts were not taken into account in calculating the mass balance, but the errors related to unscanned areas should be very small because the relative proportions of the artefacts over the entire glacier surface were minor (3.1% for summer 2015, 3.2% for 2015-16, 3.6% for summer 2016, and 4.6% for 2016-17, Fig. 5).

- P9, L13: The given errors can be listed with 2 decimal places to reflect appropriate level of certainty.
Reply: Agree! Now mass balance and uncertainty values with 2 positions after decimal were written.

- P9, L35: You should write clearly here that the value you cited indicates point mass balance.
Reply: Now revised as "…found an uncertainty of ±0.2 m w.e. a$^{-1}$ for point mass balance. Beedle et al. (2014) suggested an error of point mass balance to be about ±0.1 m w.e. a$^{-1}$ for accumulation-area measurements."

- P10, L4: What the meaning of sampling sites? Please rephrase to be more specific
Reply: Here sampling sites mean ablation stakes and snow pits (if firn exists). Now the sentence was revised as "…the number of ablation stakes and snow pits (if firn exists)…"

- P10, L4: You mean something was given in Table 4? Add some sentences to be clear.
Reply: Now revised as "Resulting values of $\sigma_{glac}$ are listed in Table 4."

5 Results
- P10, L23: Replace "clear" by "clearer"
Reply: Done.

- P10, L29: Correct "Figs"
Reply: Done.

- P10, L29-30: I know debris cover on a glacier may alleviate ablation when the debris thickness exceeds a certainty value. But your argument explaining the phenomena is quite speculative. Please support your opinion by some semi-quantitative or quantitative data.

Reply: Thanks for the constructive comments. We agree that debris cover on a glacier may

alleviate ablation when the debris thickness exceeds a certainty value. Actually, the relative proportion of debris-covered area is very small from our field observation and does not influence the calculation of glaciological mass balance. Therefore, we have not measured surface ablation of debris-cover area and can only give some qualitative explanation.

- P10, L33: Use "with" instead of "by"
Reply: Done.

- P10, L33: Please correct "Figs"
Reply: Corrected.

- P11, L2: Again: please correct "Figs"
Reply: Corrected.

- P11, L7: Rephrase "…all of the four investigated periods"
Reply: Done.

- P11, L9: It makes no sense that the value is rounded to three decimal places, please change everywhere else in the manuscript.
Reply: We have changed accordingly.

- P11, L11: Add "compared to the corresponding values of EB" after "…more negative"
Reply: Now added as suggested.

- P11, L16: Fig. 5 instead of Fig 7?
Reply: Yes! We have changed accordingly.

- P11, L17: Replace "sites" with "ablation stakes"
Reply: Done.

6 Discussion
- P12, L30: Add "of each scan positions"
Reply: Done.

- P12, L30: Fig. 4 instead of Fig. 3?
Reply: Yes! We have changed accordingly.

- P12, L32: Replace "are" with "is"
Reply: Replaced

- P13, L9: I know the number and location of ablation stakes vary from year to year as stakes melt out and sink. Please give a specific period for the average value.
Reply: Now the sentence was revised as "the average density is about 28 stakes $km^{-2}$ from 2015 to 2017".

- P13, L11: Delete "in"

Reply: Done.

- P13, L13; How did you decide the annual discharge? I guess you use the mean value here, can you calculate the internal and basal ablation using the measured data of each year?

Reply: Here $Q_m$ is mean annual discharge of glacier melting, which was determined by using the cumulative measured surface ablation over the two years. Now we checked the measured glacier surface ablation and estimated the value of $Q_m$ to be about $1.4 \times 10^9$. And then internal ablation was recalculated.

Now the paragraph was revised as:

Thus the TLS device yields accurate geodetic results and the quality of the glaciological mass balances is also very good. Nonetheless, the glaciological method cannot measure internal and basal mass balances, but these processes are implicitly captured by the repeated geodetic surveys. We need to provide a rough estimate of internal and basal mass balances of UG1 to detect their contributions to the differences between glaciological and geodetic mass balances. UG1 is a cold glacier, and its internal ablation ($B_{pe}$) is weak (Huang, 1999; Albrecht et al., 2000), mainly because of the released potential energy of descending water:

$$B_{pe} = \frac{Q_m g}{L_f \bar{s} \rho_{water}} \cdot \frac{\bar{h}_{ELA} - h_{term}}{2}, \tag{11}$$

where $Q_m$ denotes annual discharge of flowing water, g is the gravitational acceleration, $L_f$ is the latent heat of fusion, $\bar{h}_{ELA}$ and $h_{term}$ are average equilibrium-line altitude (ELA) (4152 m) and the altitude of the glacier terminus (3775 m), respectively, $\bar{s}$ is the average glacier area between 2015 and 2017. The cumulative measured glacier surface ablation over the two years was used to determine annual discharge and the value of $Q_m$ was estimated to be about $1.4 \times 10^9$. A calculation of $B_{pe} = -0.005$ m w.e. a$^{-1}$ is made.

Basal ablation from geothermal heating ($B_{gt}$) was evaluated using

$$B_{gt} = \frac{qt}{L_f \rho_{water}}, \tag{12}$$

where $q = 0.059$ W m$^{-2}$ is the geothermal heat flux (Huang, 1999), $t$ is the mass-balance period; here we primarily consider annual scale and basal ablation was estimated to be about 0.005 m w.e. a$^{-1}$. The calculated internal and basal ablation totaled -0.01 m w.e. a$^{-1}$.

We assessed internal accumulation dominated by refreezing percolating water in the cold interior of the glacier as well as the freezing of water in cold snow and firn following Zemp et al. (2010), who assumed that internal accumulation was 4% of the winter mass balance, and the resulting value was about 0.01 m w.e. a$^{-1}$ in this study. Finally the total value of internal and basal mass balances was closed to zero, which is far less than the difference ($\Delta B$)

between the two methods. This suggests that the contribution of annual internal and basal processes is negligible and does not affect the differences between the two methods.

- P13, L32: I would delete the first sentence in this paragraph as it had appeared in the introduction.
Reply: Done.

- P14, L7-9: Can you quantify the influences of unscanned areas?
Reply: Thanks for the good comments; we have explained there-in-before. The errors related to unscanned areas should be very small because the relative proportions of the artefacts over the entire glacier surface were minor (3.1% for summer 2015, 3.2% for 2015-16, 3.6% for summer 2016, and 4.6% for 2016-17).

- P14, L10: Rephrase "a discrepancy in mass balance elevation distributions of WB was observed at…"
Reply: Done.

- P14, L11: Replace "takes" with "take"
Reply: Done.

- P14, L33: The discussion about the potential of the long-range TLS to measure glacier mass balance is very weak and not really satisfying, but I believe that there is more to say. e.g. Comparison with other technologies, such as unmanned aerial systems, terrestrial photogrammetry, and then you can discuss the advantages/disadvantages of each technology. We see some data voids; can you say something about future application of such TLS to monitor glacier evolution. I am firmly sure that artefacts will also exist for other glaciological applications. The data voids can be avoided when combining with other approaches? Density conversion is still a challenge at annual and seasonal scales, which assuredly influences the wide application. What do you advise as reduction of the density conversion? I don't really think the majority of glaciers can be measured using the TLS as some of them lie at remote locations. I would rather suggest you to select some representative glaciers (evenly distributed at different mountains, different types and areas, etc.) with easily accessible locations for the geodetic mass balance monitoring. Can you discuss something about application of TLS to monitor the representative glaciers; it would be very interesting and relevant to know additional information of those glaciers for future studies. Can you give more information about TLS-derived geodetic results to validate the distributed glacier mass-balance models; I think it is very important for future glaciological studies since its high spatiotemporal resolution and the shortage of in situ measurements.
Reply: Thanks for the constructive comments; we have added new information to discuss the potential of the long-range TLS. Please see subsection 6.4 in the revised manuscript.

[revised manuscript text omitted]

- P15, L9: I think microwave remote sensing is not an effective technology as the limited time and space resolution.
Reply: We have deleted corresponding sentences.

7 Conclusions
Cloud need to be a bit change after taking account the comments mentioned above.
Reply: Now revised and adapted accordingly.

8 Figures and tables
- Figure 2: In the caption, please add some scientific content to illustrate the figure.
Reply: Now added as suggested.

- Figure 3: Please improve the figure to obtain clear content.
Rely: Now improved accordingly.

- Figure 4: Please again improve the figure to obtain clear content.

Rely: Now improved accordingly.

- Figure 5: Please again improve the figure to obtain clear content.
Rely: Now improved accordingly.

- Figure 6: Please again improve the figure to obtain clear content.
Rely: Now improved accordingly.

- Table 3: Please hold two decimal places.
Rely: Now values with 2 positions after decimal were written.

- Table 4: Please again hold two decimal places.
Rely: Now values with 2 positions after decimal were written.

---

## Referee Report (RR1)

Page 3, line 18 – change "a case" to "an example"

Page 3, lines 22-23 – change "put forward" to "suggested"

Page 4, line 3 – change "decide" to "determine"

Page 5, line 1 – change "compare" to "compared"

Page 5, line 29 – delete "also"

Page 6, line 33 – insert "$\rho_{firn}$ is the density of firn"

Page 7, line 4 – change "t" to "the"

Page 7, line 27 – change "different" to "difference"

Page 8, line 2 – change "Accurate" to "An accurate"

Page 8, line 3. Start a new sentence where the semi-colon is.

Page 8, line 34. Delete «in the» (or do alternative correction of sentence so that it makes sense).

Page 9, lines 31-32 Change sentence to "There are additional sources of error in the glaciological measurements that lead to uncertainties in glaciological mass balance that are not easy to quantify (Dyurgerov, 2002)."

Page 9, line 32. Change beginning of sentence to "These uncertainties were classified into …."

Page 10, lines 34-36. Correct this sentence from "variations with smaller ……." So that it is comprehensible.

Page 11, line 3. Delete "During summer periods".

Page 11, line 4. Change "first" to "previous"

Page 11, line 6. Change "addations" to "addition".

Page 11, line 8. Start a new sentence at "a slight thinning area ……" and change to "An area of minor thinning ….."

Page 11, line 13. Repetition of "the beginning of" – delete one of them.

Page 11, line 14. Change "filed" to "field".

Page 13, line 19. Change "balances" to "balance".

Page 14, lines 8-9. Change "ice temperature of the glacier bed has not reached the melting point" to "the temperature at the glacier bed is below the melting point of ice".

Page 14, lines 12-13. Change this sentence to "In addition, internal melt caused by changes in potential energy due to glacier dynamics is negligible, as the glacier dynamics themselves are insignificant".

Page 14, line 19. Change "A calculation of" to "Equation (11) gives".

Page 14, line 20. Delete "is made".

Page 14, line 21. Correct spelling to "Storglaciären".

Page 14, line 22. Change "clod, continental" to "cold".

Page 14, lines 22-24. Repetition of a phrase – delete one of them.

Page 14, line 25. Change ", measurement and studies on internal accumulation of the is still blank" to "with no measurements of internal accumulation".

Page 14, line 28. Change "hardly" to "there is hardly".

Page 15, lines 3-4. Change "makes dynamic thinning of Urumqi Glacier No. 1" to "causes dynamic thinning of the glacier".

Page 15, line 4. Delete "make".

Page 15, line 12. Change "since" to "due to".

Page 15, line 20. Change to "Applying a reciprocal density conversion to the mass balance differences provides estimates of the ………."

Page 15, line 21. Change both "is" to "as".

Page 15, line 22. Change "from" to "the".

Page 15, line 23. Change "duo" to "due".

Page 15, line 26. Change "misalign" to "misalignment".

Page 16, line 2. Change "Despite" to "although".

Page 16, line 7. Change "relative" to "a".

Page 16, line 8. I assume that "data" should be "date".

Page 17, line 3. Change "stakes" to "stakes can".

Page 17, line 25. Change "has" to "have".

Page 17, line 27. Change this sentence to "Hence it should be possible to measure most glaciers using the TLS".

Page 17, line 28. Delete "of".

Page 17, line 29. Delete "number".

Page 17, line 31. Change "locate" to "are located".

Page 17, line 32. Delete "select".

Page 17, line 33. Change to "… system thus has a huge potential for glacier ……….."

Page 17, line 39. Change "annul" to "annual".

Page 17, line 40. Change sentence to "A day with little snow in the accumulation"

Page 17, line 41. Change to "… and no snow in the ablation area …"

Page 18, lines 27-28. Change "What's more" to "Further more".

---

## Author Response (AR2)

**Long-range terrestrial laser scanning measurements of summer and annual mass balances for Urumqi Glacier No.1, eastern Tien Shan, China**

Chunhai Xu et al.

We would like to thank the referee for the constructive and detailed comments. Reviewer comments are copied in normal font, and our point-by-point reply to each comment is provided after the comments and given in bold font.

**Summary**

I have reviewed the manuscript "Long-range terrestrial laser scanning measurements of summer and annual mass balances for Urumqi Glacier No.1, eastern Tien Shan, China" submitted to The Cryosphere by Xu et al. The article describes using terrestrial laser scanning to look at annual and intra-annual mass balance for Urumqi Glacier No. 1 over a two-year period, and compares the geodetic results with the glaciological mass balance.

**Reply: Thanks!**

The article presents some interesting results, but there is not much new material compared with the articles from 2017 and 2018. It is not clear (apart for the obvious reasons) why the authors are publishing these results in several papers. It is suggested that further analysis is done of the results by incorporating velocity measurements, and going into more detail of the thickening or balance at higher elevations, and thinning at lower elevations.

**Reply: Thank you for the constructive suggestions. For the published paper in 2017, we used the TLS to implement two measurements one month apart (25 April-28 May 2015) to monitor the monthly net mass balance of UG1 Urumqi Glacier No.1 at the monthly scale (25 April-28 May 2015), however the result of this paper are preliminary. What's more, we have not compared the glacier-wide mass balance between the two methods by considering many factors (e.g. density, data quality, internal and basal mass balance, glacier vertical velocity). The article published in 2018 mainly reanalysis the mass balance of Urumqi Glacier No.1 and evaluate the quality of mass balance records according to the proposal of WGMS. China contains the large number of glaciers around the world, and most of these glaciers are summer-accumulation type, now only several glaciers have discontinuous glaciological mass balance records. The aim of this study is thus to established an optimization scheme of volume-to-mass conversion to realize the calculation of TLS-derived geodetic mass changes, to investigate the possible causes of the differences between glaciological and geodetic mass balance. The potential of such long-range TLS to measure mass balance of glaciers in western China is evaluated and several main considerations for a wide application of the TLS in glaciology are put forward. So publishing results of this paper is prerequisite for a wide**

application.

Now we have added a section (6.4) to further analysis as suggested:

"6.4. Glacier vertical velocity component

Geodetic measurements of glacier surface elevation changes include glacier surface mass balance and vertical velocity component (Kaser et al., 2003; Geist et al., 2005). Vertical velocity ($w_s$) is downward (submergence) and makes dynamic thinning of Urumqi Glacier No.1 in the accumulation area, and in the ablation area, vertical velocity is upward (emergence) and makes dynamic thickening of the glacier. This dynamic process results in the general difference between the elevation-distributed mass changes stated above (Fig. 8). To discuss the influences of vertical velocity component, here $w_s$ depends on the kinematic boundary condition at Urumqi Glacier No.1 surface as basal sliding and bed deformation of the glacier are negligible (cf. Petterson et al., 2007; Cuffey and Paterson 2010)

$$\dot{h} = \frac{\dot{b}}{\rho} + w_s - u_s \frac{\partial S}{\partial x} - v_s \frac{\partial S}{\partial y} \tag{13}$$

in which $\dot{h}$ is the rate of glacier surface elevation changes, $u_s$ and $v_s$ are the components of horizontal ice velocity at the glacier surface s, respectively (Cuffey and Paterson 2010). We neglect the advection of the glacier surface topography induced by horizontal ice flux since the low reduced horizontal velocity (Wang et al., 2017) and short time span of this study. Then changes in $\dot{h}$ equals the sum of $\frac{\dot{b}}{\rho}$ and $w_s$, can be expressed as (Beedle et al., 2014)

$$\dot{h}\rho - \dot{b} = w_s\rho \tag{14}$$

Glacier dynamic thinning and thickening can be detected by subtracting the glaciological mass balances from the geodetic ones. For most of the study periods, positive difference values (thickening) dominate in the lower elevations, especially the glacier tongue, and negative difference values (thinning) mainly occur in the higher parts (Fig. 7a, d, g and j). Positive values across the east branch in summer

2015 may be related to different survey dates between the geodetic and glaciological methods.

Now applying reciprocal density conversion to the mass balance differences estimate the submergence and emergence velocities. Here we defined the term submergence is negative vertical velocity and emergence is positive vertical velocity. Variation tendency of the estimated velocities at ablation stakes were found to match from in situ measured ones, especially for east branch (Fig. 8). Relative bigger differences of west branch were detected in the mass balance years 2016-17 (Fig. 8b and d), which may duo to an avalanche in the upper part during the summer 2017. The firn basin terrain of west branch is very steep and is adverse to mass accumulation, which can also be validated in terms of TLS-derived glacier surface elevation changes (Fig. 4g). Thus pronounced misalign of mass balance elevation distribution curves between the two methods occurred. Considering the errors of estimate and in situ measurements, submergence and emergence velocities can be estimated using the TLS-derived DEMs and glaciological mass balance. The difference in mass balance elevation distribution can be largely explained by glacier dynamic thinning at higher elevations and dynamic thickening at lower elevations.

[Figure]

Figure 8. Comparison between estimated and in situ measured vertical velocity for the mass

balance year 2015-16 and 2016-17; the letters represent ablation stakes. Note than the summer periods and stakes in the higher elevations were not selected for comparisons due to snow cover reduced the quality of in situ measured vertical velocity.

In fact, the vertical velocity of Urumqi Glacier No.1 is small (Fig. 8), we now discuss the errors of glacier surface elevation changes versus dynamic thinning and thickening. Differences in glacier surface elevation changes derived from the TLS and glaciological measurements were close to zero for the vast majority of the ablation stakes, and corresponding errors in the differences were mostly larger than the difference themselves (Fig. 9). Compared with the errors of measurements, dynamic thinning and thickening of the glacier were minor and negligible.   So Riegl VZ®-6000 TLS can be considered as an effective tool to measure the mass balance of Urumqi Glacier No.1.

[Figure]

Figure 9. Changing differences between TLS-derived ($\Delta h_{\text{TLS}}$) and glaciological ($\Delta h_{\text{glac}}$) annual and summer surface elevation changes at individual stakes versus normalized glacier elevation range of east branch and west for the four investigated periods. Note than dash vertical lines represents the uncertainty ranges ($\sqrt{\sigma^2_{\Delta h\text{TLS}} + (\sigma_a^{\text{ice}})^2 + (\sigma_a^{\text{firn}})^2}$) of the changing differences ($\Delta h_{\text{TLS}} - \Delta h_{\text{glac}}$), and grey quadrants indicate theoretical areas for Urumqi Glacier No.1 in equilibrium."

Generally, the article is difficult to read for several reasons. The English needs to be considerably improved, there is an overabundance of acronyms and the order in which topics are dealt with is not always logical.

**Reply: We have already corrected syntactic problems and improved English in the whole manuscript accordingly. For acronyms, we have written out many, such as: "ICP" is replaced by "iterative closest point"; "DMS" is replaced by "Daxigou Meteorological Station", UG1, EB and WB are written out Urumqi Glacier No.1, east**

**branch and west branch, respectively. Organizational order of the manuscript has also been changed.**

Acronyms should be avoided where possible. E.g. ICP is used only twice, so should be written out. Write out what the acronym means on first use, even for terms such as "w.e.". DMS should be replaced by "the meteorological station" or "the met. station". Use of this acronym every time the met station is referred to is confusing.

**Reply: Thanks for the carful and detailed comments. Now we have already checked everywhere else in the manuscript and write out acronyms. ICP means iterative closest point and is written out accordingly. "w.e." indicates "water equivalent" and now we write out it means on first use. DMS has been replaced by the full name as suggested.**

**Comments on the text and grammar**

p. 1, line15-16. Change to "…. well-suited for repeated glacier mapping, and thus determination of annual and seasonal geodetic mass balance."

**Reply: Now corrected accordingly**

line 18. Change to "for two consecutive mass balance years ……."

**Reply: We added as suggested**

Line 24. Change "satisfying" to "satisfactory"

**Reply: Corrected**

Line 28. Change "function known as" to "concept of"

**Reply: Corrected**

Line 34. Change "Ongoing" to "Continuous"

**Reply: Corrected**

p.2 line 9. The geodetic method doesn't "measure" all mass balance processes as such. Change the sentence to "The method includes all processes that affect the surface ………."

**Reply: We fully agree. The sentence has been rewritten as suggested.**

Line 15. Change "burgeoning" to "emerging"

**Reply: Corrected**

Line 17. Change to "to calculate geodetic mass balance and changes in glacier volume …"

**Reply: Corrected**

Lines 17-19. It is unnecessary to have to many references. Several of these had no new techniques, merely applying the same technique to different areas. Also, add the first studies of intra-annual changes in mass balance from laser scanning, Pellikka, P. and W.G. Rees, eds. 2010. Remote sensing of glaciers: techniques for topographic, spatial, and thematic mapping of glaciers. Boca Raton, FL, CRC Press/Taylor & Francis. 330pp. ISBN-10: 0-415401-66-6, ISBN-13: 978-0- 415-40166-1 (or Vetter et al, 2009 article) and Geist et al (2005), Investigations on Intra-annual elevation changes using multi-temporal airborne laser scanning data: Case study Engabreen, Norway.

**Reply: As suggested by the referee, we have deleted some of the listed references and cited the first studies of intra-annual changes in mass balance from laser scanning mentioned above.**

Line 19. Replace "advantageous" with "effective". Replace "wide" with "extensive".

**Reply: Done**

Lines 19-21 require more explanation – "what is meant by "the difficulty of studying small-scale processes"? Small-scale processes are not the focus here. In addition, why is the presence of rock outcrops a problem for airborne observations, rather than ground-based? Usually the opposite would be assumed.

**Reply: Thanks for the detailed reviews. For the first sentence: we want to clarify TLS system can more easily capture glacier changes with high time resolution than ALS since the high cost of aircraft. For the second sentence: most ALS instruments have limited operating flight altitude, such as the novel RIEGL VQ-780i ALS has a maximum fighting altitude of 5600 m a.s.l., Leica ALS70 with a maximum altitude of 5000 m a.s.l., Optech ALTM Gemini with a maximum altitude of 5000 m a.s.l., etc. However, the maximum altitude of most glaciers (78.6% of the total number) in western China exceeds 5000 m a.s.l. according to the second Chinese glacier inventory, hence great topographic relief and high altitude of rock outcrops around glaciers usually increase the difficulty of aircraft flight. In order to be more reasonable, these sentences were rephrased as follows:**
**"but the difficulty of studying glacial changing processes with high temporal resolution since the high costs of ALS and the presence of great topographic relief and high-altitude rock outcrops around glaciers reduce the capacity of observations by aircraft as most ALS instruments have limited operating flight altitude, so we need ground-based surveys (Piermattei et al., 2015)."**

Line 22. Change "evolutions of" to "changes in".

**Reply: Corrected**

Line 23. Delete "reference glaciers in particular", as this is irrelevant. Change "-resolution" to "high.resolution".

**Reply: Done**

Lines 25-26. From "Being" change to "The scanner is a Laser Class 3B, with laser wavelength in the near-infrared (~1064 nm), and thus well.suited for measuring snow- and ice-covered terrain in….."

**Reply: Done**

Line 27. "Some recent studies" – is it some or one? The reference is to Gabbud et al, 2015, which describes a study on one glacier.

**Reply: Now we rephrased the sentence as "One study has…..."**

Lines 28-32. From "however", change to "however, only the middle and lower elevations were measured as the glacier is relatively big. Another study reports the performance of the Riegl VZ®-6000 in monitoring 30 the mass balance of five glaciers in the European Alps; the surface terrain of each glacier can be almost entirely detected using one scan position since these glaciers are very small and have steep terrain (Fischer et al., 2016). For medium-sized and large glaciers with flat terrain, however, a single scan position cannot capture the whole glacier surface."

**Reply: These sentences were rephrased as suggested.**

Line 35. Delete "-term"

**Reply: Done**

Line 36. Delete "Riegl VZ® 6000"

**Reply: Done**

p.3. Line 1. Break the sentence up, e.g. full-stop after results, and start the next sentence "An accurate …."

**Reply: Now changed accordingly**

Line 2. Change "received attention" to "been performed"

**Reply: Corrected**

Line 3. Was the study done at a monthly scale, or were two measurements made one month apart to get net mass balance over one month? This needs to be clearer.

**Reply: Thanks for the detailed review. Actually, the sentence wants to mean that two measurements made one month apart to get net mass balance over one month. Now the sentence was rephrased to be clearer:**

**"Our previous study has used the TLS to implement two measurements one month apart (25 April-28 May 2015) to get monthly net mass balance of UG1"**

Lines 6-7. "besides we only considered snow/firn densities in the geodetic mass balance calculations" etc. – this sentence doesn't make sense.

**Reply: Now the sentence was rewritten as:**
**"besides we only considered snow/firn densities in the determination of a density conversion, which was used to convert monthly volume change to geodetic mass changes, as an abundance of fresh snow covered the entire glacier surface at the time of the TLS surveys (Xu et al., 2017)."**

Line 8. Change "mass balance" to "mass changes".

**Reply: Corrected**

Line 9. What is meant by "Meteorological influences on the elevation changes" from Huss (2013)? Is this referring to changing mass balance gradients? Generally, this whole paragraph down to line 24 on page 3 needs to be rewritten. Bader (1954) is referenced later in the text, but could be introduced here, in lines 13-14.

**Reply: Yes, this refers to changing mass balance gradients. Bader (1954) is referred in lines 13-14. We now rewrite the whole paragraph down to line 24 as suggested:**
**Urumqi Glacier No.1 (hereafter known as UG1) has the most detailed annual and seasonal surface mass balance measurements in China. It is also one of the reference glaciers in the World Glacier Monitoring Service (WGMS) network due to its long data series, important location and significant local water supply (Li et al., 2011; Zemp et al., 2009). TLS surveys of UG1 were initiated on 25 April 2015 for four scan positions (Fig. 1a), and the subsequent measurements were nearly coincident with days of glaciological mass-balance measurements. Multi-temporal high-resolution and -precision TLS-derived DEMs are therefore available. To date, comparison of glaciological and geodetic mass balances of UG1 was reported for the period 1981-2009 at intervals of several years (Wang et al., 2014) and for the period 1981-2015 (Xu et al., 2018), but these studies used a series of low-quality topographic maps to calculate sub-decadal and decadal geodetic results. An accurate reanalysis of seasonal and annual glaciological mass balance of UG1 using high-resolution and -precision DEMs has not been performed. Our previous study has used the TLS to implement two measurements one month apart (25 April-28 May 2015) to get monthly net mass balance of UG1, whereas we simply compared glaciological and TLS-derived geodetic elevation changes of individual stakes, whether agreement between the glaciological and TLS-derived glacier-wide mass balance was pending, potential of such technology applied in seasonal and annual glacier mass-balance measurements in western China had not been discussed; besides we only considered snow/firn densities in the determination of a density conversion, which was used to convert monthly volume change to geodetic mass**

changes, as an abundance of fresh snow covered the entire glacier surface at the time of the TLS surveys (Xu et al., 2017). In fact, the volume-to-mass conversion becomes more challenging over short time periods as meteorological factors change mass balance gradients (Huss, 2013). Several recent studies have used an area-weighting method to calculate the annual density conversion by classifying a glacier surface into bare ice and firn (Fischer et al., 2016; Klug et al., 2018). But the volume changes in ice and firn/snow usually take place at the same vertical layer for summer-accumulation-type glaciers (accumulation and ablation take part simultaneously in summer months) from our field observations, it is therefore inappropriate for this study to adopt the area-weighting method. Besides, compaction and metamorphosis imply a shift in the vertical firn profile as well as changes in firn thickness and density (Cuffey and Paterson, 2010; Ligtenberg et al., 2011),so assuming no change occurs in the vertical firn density profile over time in the accumulation area is unrealistic (Bader, 1954).

This study takes UG1 as a case and describes the use of the TLS to monitor annual and seasonal geodetic mass balances for two consecutive mass balance years (2015-17). The aim of this study is thus to established an optimization scheme of volume-to-mass conversion to realize the calculation of TLS-derived mass changes, to investigate the potential of such long-range TLS to measure mass balance of glaciers in western China and to put forward some main considerations for a wide application of the TLS.

Line 17. Change "consecutive years" to "consecutive mass balance years".

**Reply: Corrected**

Line 30. Why "accelerated" recessions? Do you mean just recessions?

**Reply: Yes we mean recessions. Three time periods with different melting rates have been identified according the slope of the accumulative mass balance curve (Figure R1). This indicates that there were two accelerations in the melting processes. In order to clarify clearly, we replace "recessions" with "mass loss".**

[Figure]

**Figure R1 Cumulative mass balance of UG1 with linear regressions (Li et al., 2011)**

Line 30. Delete "was" – this was a natural occurrence.

**Reply: Corrected**

Line 31. Why "enhanced"? Do you mean "increased"? If increased, over which period?

**Reply: The air temperature rises during melting season, the ice temperature augment of the glacier and the albedo reduction on the glacier surface induce enhanced melting (Li et al., 2011). Here "enhanced" is same as "increased". But the increased melting only observed in recent 50 years, we cannot subjectively attribute the separation to the observed result. So we delete "due to enhanced melting" to write as precise as possible.**

p. 4, line 3. Change "accumulation rate is quicker" to "accumulation is higher".

**Reply: Corrected**

Line 17. Change "-precision" to "high-precision".

**Reply: Corrected**

Lines 24-25. Change to "point. The four scan positions were surveyed using real-time kinematic …….."

**Reply: Corrected**

Line 26. Change "facilitate" to "give".

**Reply: Corrected**

Line 27. Change "RTK surveys" to "survey".

**Reply: Corrected**

Line 32. Delete "As to UG1"

**Reply: Deleted**

Line 35 to p. 5 line 1 Change "no less than" to "at least"

**Reply: Corrected**

p. 5. Line 19. "advantageous" – compared with what?

**Reply: "the direct georeferencing technique in TLS using global navigation satellite systems (GNSS) is advantageous compare with total stations and the inclination sensors (Paffenholz et al., 2010; Mukupa et al., 2016)"**

**Reply: Done**

Line 4 – insert «descriptive» in front of «free water content». This is in the original article, otherwise you are suggesting that free water content can be calculated merely by digging a snowpit.

**Reply: We agree and insert accordingly**

Line 10. Change to "………….observed using stakes and snow pits, since 1959 ……..".

**Reply: Corrected**

Line 15. Change "where snow has accumulated" to "in the accumulation area".

**Reply: snow pits were not always dug in the accumulation area, fresh snow usually covers the whole glacier at the begin of the ablation season, we should dig snow pits at each ablation stakes**

Line 27. After respectively, insert "and the different between these two, i.e."

**Reply: Corrected**

p. 8, line 4. "probably led to an overestimate of the glacier extent" – when?

**Reply: Now we revised the sentence as "Fresh snow cover probably led to an overestimate of glacier extent at the beginning of the ablation season"**

Line 11. Change "assessments" to "assessment".

**Reply: Corrected**

Line 20. Why "finish"? replace with "perform"?

**Reply: We agree "perform" is better than "finish", now replaced accordingly**

Line 22. Change "stabilized" to "established".

**Reply: Corrected**

p. 9, line 1. "quantitatively". Should this be "qualitatively"? Otherwise the sentence doesn't make sense.

**Reply: Now changed accordingly**

Lines 5-6 – This sentence regarding the supraglacial river doesn't make sense and needs to be rewritten.

**Reply: We have rewritten as suggested**

Line 34. Change beginning of this paragraph to "There are additional sources of error for ......."

**Reply: Now changed accordingly**

P. 10, line 11. Change "glaciers" to "glacier".

**Reply: Corrected**

Lines 11-12 – the figures given here from Andreassen et al (2016) differ from the values given in the article.

**Reply: We have checked the reference Andreassen et al (2016) and corrected the figures. Now we revised as:**
"For Nigardsbreen (Norway) glacier, Andreassen et al. (2016) calculated a point measurement of ±0.25 m w.e. a$^{-1}$ by summing false determination of the summer surface (±15 m w.e. a$^{-1}$), subsidence of stakes (0.20 m w.e. a$^{-1}$), errors in snow (0.05 m w.e. a$^{-1}$) and firn (0.02 m w.e. a$^{-1}$) density measurements."

Line 34. Why "remarkable"? What is remarkable about it? Or do you mean significant?

**Reply: here, relative bigger difference (ΔB) between glaciological and geodetic mass balances was seen. We now rephrased the sentences in order to be clear:**

**"In 2016-17, the difference ($\Delta B = B_{\mathrm{glac}} - B_{\mathrm{geod}}$) in glacier-wide mass balances of**

**UG1 between the glaciological and geodetic methods was close to zero. Significant differences between the two mehods were detected in summer 2015 for UG1 and EB, with $\Delta B$ = -0.24 m w.e. and $\Delta B$ = -0.27 m w.e., respectively. In other three periods, the differences were much less the uncertainties of $\Delta B$, which were**

**calculated based on the law of error propagation ($\pm\sqrt{\sigma_{geod}^2 + \sigma_{glac}^2}$)."**

p. 12. Line 6. Replace "satisfying" by "satisfactory".

**Reply: Corrected**

Line 8. Replace "know" by "give"

**Reply: Corrected**

Line 10. Change "at the steep elevations" to "on steep slopes".

**Reply: Corrected**

Lines 13-15. These two sentences need to be further clarified. Where are the annual vertical ice velocities reported? The geodetic results were more positive in lower-elevation regions – does this not contradict results for the Western Branch given on page 11?

**Reply: Thanks for the careful review. Wang et al (2017) has not studied the vertical ice velocities, an early study reported the annual vertical ice velocities (Sun et al., 1985). The observed annual vertical ice velocities were in the range of -1.14--0.07 m a$^{-1}$ (with a mean value of -0.48 m a$^{-1}$) and -1.17~-0.08 m a$^{-1}$ (with a mean value of -0.49 m a$^{-1}$) for mass balance year 2015-16 and 2016-17, respectively (TGS, 2016, 2018). For UG1, both the glaciological and geodetic mass balances were negative in lower elevations and slight positive in higher parts. Here we want to write the geodetic results were more positive in lower-elevation regions compare with the glaciological mass balance. Now we added analysis about vertical velocity, related contents in this part were deleted.**

**Compared to the mass-balance year 2015-16, areas of clearer increase were observed in the upper eastern parts of EB in the mass-balance year 2016-17, but ice losses in the lower-elevation parts and glacier thickening in the upper reaches of WB were greater in the first mass balance year (Figs. 4c, g). . During summer periods, surface lowering in summer 2015 mainly occurred in the ablation areas of EB (Fig. 5a), glacier surface ablation was significantly greater in summer 2016 than in the first summer (Fig. 4e). For a completed mass balance year 2015-16, glacier thinning areas and values in summer were obviously bigger than the whole year, which may be related to fresh snow covered the glacier at the beginning of ablation season. In addition, there were some curves of pronounced glacier surface lowering in the ablation areas during summer periods, which were related to supraglacial river (Fig. 1c, d), a slight thinning area is detected at the lower lift (northerly) edge of EB, which may be associated with debris cover (Fig. 1c).**

Section 5.2. This needs more explanation for why annual density conversions higher

than summer. The converse would be expected.

**Reply: Due to the mass changes in ice and firn/snow occur at the same vertical profile for Urumqi Glacier No.1, we should consider three-dimensional changes of ice and firn/snow bodies, we us a volume-weighting method to determine density conversion according to the principle of glaciological mass balance calculations:**

$$\rho_i = \frac{\Delta h_{\mathrm{ice}} \cdot \rho_{\mathrm{ice}} + \Delta h_{\mathrm{firn}} \cdot \rho_{\mathrm{firn}}}{\Delta h_{\mathrm{ice}} + \Delta h_{\mathrm{firn}}}$$

**where $\rho_{\mathrm{ice}}$= 900 kg m$^3$ is glacier ice density, $\rho_{\mathrm{firn}}$ is firn/snow density, $\Delta h_{\mathrm{ice}}$ and $\Delta h_{\mathrm{firn}}$ are the changes in ice and firn/snow thickness, determined from glaciological mass-balance measurements.**

**Urumqi Glacier No. 1 is a summer-accumulation-type glacier, annul mass balance of the glacier is defined from the previous September 1 to the next August 31, summer runs from the beginning of May to early September each year (Liu et al., 1997). Large amounts of fresh snow or firn cover the glacier surface at the beginning of May (Figure R2), but glacier bare ice in the ablation area exposes and snow at the accumulation area is thinner on early September (Figure R3). So single-point density conversion $\rho_i$ on the summer months is smaller than that of annual values.**

[Figure]

**Figure R2 fresh snow cover the glacier surface at the beginning of May (the photo takes on 29 April 2017)**

[Figure]

**Figure R3 The glacier bare ice in the ablation area exposes and snow at the**

**accumulation area is thinner on early September (the photo takes on 1 September 2016)**

**Now we more explanation about annual density conversions higher than summer and rewrite as follows:**
**"The thicker snow and firn covered the whole glacier surface at the beginning of the beginning of May each year and the ablation area was bare ice or covered by a thin snow layer at the end of the ablation season according to filed observations (Liu et al., 1997; Xie and Liu, 2010), so the changes in ice and firn/snow thickness are observed during the summer months. However, firn and snow densities are far smaller than glacier ice density, these result in annul single-point density conversion $\rho_i$ is bigger and the glacier-wide annual density conversions were accordingly higher than the summer ones (Table 3). "**

Line 13. Change to "covered by a thin snow layer at the end ……."

**Reply: Corrected**

P. 13, line 2. Change "locate" to "be located"

**Reply: Corrected**

Line 28. Change "glaciers" to "studies"

**Reply: Corrected**

P. 14, Lines 1-18. It would make more sense to move the discussion of internal accumulation to where internal ablation is discussed. It is confusing to state that UG1 is a cold glacier, then to jump straight into internal ablation.

**Reply: Thanks for the suggestions, now moved accordingly. Urumqi Glacier No.1 is a small and cold glacier, bottom sliding and bed deformation of the glacier is considerable to be negligible since temperature of the glacier bed has not reached the melting point, so the ice velocity of the glacier is very small (Huang, 1999; Wang et la., 2017). Previous studies have concluded that internal ablation of poly-thermal glaciers is negligible as the ice motion is small (c.f. Albrecht et al., 2000, Zemp et al., 2010). In addition, Urumqi Glacier has low reduced dynamics and hardly any subglacial water systems. Thus, we can consider internal ablation of the glacier is weak. In order to be clearer, corresponding paragraph was rewritten as follows:**
**"UG1 is a small and cold glacier, bottom sliding and bed deformation of the glacier is negligible since ice temperature of the glacier bed has not reached the melting point, so the glacier has low ice velocity and dynamics, and hardly any subglacial water systems (Huang, 1999; Xie and Liu, 2010; Wang et a., 2017). Previous studies have suggested that internal ablation of poly-thermal glaciers is negligible as the ice motion is small (e.g. Albrecht et al., 2000; Zemp et al., 2010).Besides internal**

melt due to the potential energy of glacier dynamics is negligible since the low reduced dynamics. Thus, internal ablation of UG1 is weak and mainly comes from the released potential energy of descending water"

**Figures**

Figure 1 – Positions of stakes and density pits are difficult to see and should be made clearer.

**Reply: Now made clearer accordingly, see figure below:**

[Figure]

Figure 2 – this figure and reference in text are unnecessary, a reference to Litchi et al (2005) is sufficient.

**Reply: We deleted the figure and referred Litchi et al (2005) as suggested.**

Figure 3 – The glacier boundaries for different years needs to be clearer. Very hard to see the difference now. The (b) and (c) on the figures is hard to read.

**Reply: Now made clearer accordingly, see figure below:**

[Figure]

Figure 5 – The green polygons are very hard to see. In caption, change "(the unit is m)" to "(in m)"

**Reply: Now the figure was made clearer accordingly. In caption, we changed "(the unit is m)" to "(in m)". See figure below:**

[Figure]

Figure 6 – The caption refers to black lines where are these? Also, these are meant to be the same as the corresponding boundary of figure 3, but this figure shows several glacier boundaries. "artefact" is not the correct terminology, as it refers to something that is the result of the measurement technique or experiment, not merely missing data. "Areas" would be a better term than artefacts.

**Reply: Thanks for the comments, here black lines are glacier boundaries. Now we revised as: "Here, glacier boundaries in (a) and (b) are same as boundary 2015 of fugure3, glacier boundaries in (c), (d), (e) and (f) are same as boundary 2016 of**

**fugure3, glacier boundaries in (g) and (h) are same as boundary 2017 of fugure3."**
**We agree "artefact" is not the correct terminology and now replaced "artefacts"**
**with "areas".**

Figure 8 – Elevations in left-hand column are difficult to read – need to use a bigger
font. The text in the middle and right-hand columns is also difficult to read and needs
to be bigger.

**Reply: We have used a bigger font to make the figure easy to read.**

Figure 9 – change "temperature" to "air temperature"

**Reply: Corrected.**

Figure 11 – this map is unnecessary, as all it shows is spatial distribution of glaciers
with suitable area and slope. If for some reason this figure is retained, change area
cut-off to 1.5 km2; meaningless to use 1.555 km2.

**Reply: Thanks! This map shows application potentiality of the long-range TLS for**
**glacier mass-balance monitoring in China, now we have selected some benchmark**
**glaciers among these appropriate glaciers to implement densely TLS measurements,**
**in the next steps, we will selected more for TLS measurements to enrich the**
**observational data. Hence, we retained this reasons. The area threshold with 1**
**position after decimal was given.**

Table 2. Change "are calculated based on the elevation changes over stable terrain"
to "are measures of error derived by calculating elevation changes from TLS over
stable terrain (off-glacier)". In the header columns give the terms not symbols, e.g.
Error or StdDev, number of points, mean (stable terrain) and standard error (stable
terrain). Give the periods in the caption rather than under the table.

**Reply: Now changed as suggested. We have given related terms in the header**
**columns and the observed periods in the caption, see the table below:**

[revised manuscript text omitted]
 \text{TLS}}} = \sigma^2_{\Delta h \text{TLS}} \cdot \frac{1}{5} \cdot \frac{S_{\text{cor}}}{S}, \tag{7}$$

where $\sigma_{\Delta h \text{TLS}}$ denotes the standard deviation of TLS-derived elevation changes over stable terrain. $S_{\text{cor}}$ is spatially correlated area. Given the high density (> 1 point m$^{-2}$) of the TLS data, we can probably assume that the number of independent items is about the number of glacier pixels (cf. Joerg et al., 2012). Here we therefore assume $S_{\text{cor}} = S$. This leads to calculated values of $\sigma_{\overline{\Delta h \text{TLS}}}$ range from ±0.16 to ±0.25 m (Table 2).

Uncertainties related to the density conversion for a single point ($\sigma_{\rho i}$) were calculated as

$$\sigma_{\rho i} = \frac{\Delta h_{ice} \cdot \sigma_{\rho \text{ice}} + \Delta h_{firn} \cdot \sigma_{\rho firn}}{\Delta h_{ice} + \Delta h_{firn}}, \tag{8}$$

where $\sigma_{\rho \text{ice}}$ and $\sigma_{firn}$ are uncertainties of ice and firn densities, which were assumed to be ±17 and ±50 kg m$^{-3}$, respectively, following Klug et al. (2018). We then extrapolated single-point values to glacier-wide uncertainties ($\sigma_\rho$) using the interpolation method on the ArcMAP 10.2 platform (Table 3). According to Huss et al. (2009), the uncertainties of the geodetic mass balance ($\sigma_{\text{geod}}$) can be estimated using

$$\sigma_{\text{geod}} = \pm \sqrt{\left(\overline{\Delta h \text{TLS}} \cdot \sigma_\rho\right)^2 + (\rho \cdot \sigma_{\overline{\Delta h \text{TLS}}})^2}, \tag{9}$$

[revised manuscript text omitted]

---

## Author Response (AR3)

**Long-range terrestrial laser scanning measurements of annual and intra-annual mass balances for Urumqi Glacier No.1, eastern Tien Shan, China**

Dear editor,

We would like to thank you and the reviewer for your careful review. We have carefully revised the manuscript by taking the stylistic comments into consideration. The comments are copied in normal font, and our point-by-point reply to each comment is provided after the comments and given in bold font. A marked-up manuscript version showing the changes is attached to our reply to the comments.

Once again, thank you for your consideration of our revised manuscript for publication in TC.

Best regards,

Chunhai Xu et al.

**Reply to comments**

**Summary**

First of all, I apologise for the long review process. This was also partly due to the fact that not all comments were adequately addressed in the first revision rounds making the re-reviews quite laborious, but especially because there was significant overlap with previous publications of similar authors. This was also stated in my earlier decisions. It was therefore very difficult to find suitable reviewers. Having stated this I want to emphasise again that single article about this glacier, instead of this and the two most recent would have been much preferred. However, as the study has now much improved (esp. the figures as well as much, although not all, of the flow of the discussion and the language) and contains interesting and relevant information I am willing to accept the manuscript after the more stylistic comments of the reviewer (He/she did really a good job which should be acknowledged.) were addressed.
**Reply: Thanks! We now read the manuscript carefully and corrected errors accordingly, including the stylistic problems and references. Additionally, all of the figures had been checked and some of them have been improved again. Please see the marked-up manuscript version.**

**Comments on the text and grammar**

Page 3, line 18 – change "a case" to "an example"
**Reply: Corrected.**

Page 3, lines 22-23 – change "put forward" to "suggested"
**Reply: Corrected.**

Page 4, line 3 – change "decide" to "determine"
**Reply: Corrected.**

Page 5, line 1 – change "compare" to "compared"
**Reply: I guess you mean the word in line 19 and changed as suggest**

Page 5, line 29 – delete "also"
**Reply: Now deleted accordingly.**

Page 6, line 33 – insert "ρfirn is the density of firn"
**Reply: Added accordingly.**

Page 7, line 4 – change "t" to "the"
**Reply: Corrected.**

Page 7, line 27 – change "different" to "difference"
**Reply: Corrected.**

Page 8, line 2 – change "Accurate" to "An accurate"
**Reply: Corrected.**

Page 8, line 3. Start a new sentence where the semi-colon is.
**Reply: Done.**

Page 8, line 34. Delete «in the» (or do alternative correction of sentence so that it makes sense).
**Reply: Now deleted accordingly.**

Page 9, lines 31-32 Change sentence to "There are additional sources of error in the glaciological measurements that lead to uncertainties in glaciological mass balance that are not easy to quantify (Dyurgerov, 2002)."
**Reply: Corrected.**

Page 9, line 32. Change beginning of sentence to "These uncertainties were classified into …."
**Reply: Corrected.**

Page 10, lines 34-36. Correct this sentence from "variations with smaller ……." So that it is comprehensible.
**Reply: Now revised as "elevation changes are more positive and show smaller lowering to pronounced thickening in the upper-elevation part except for west branch in the mass-balance year 2016-17"**

Page 11, line 3. Delete "During summer periods".
**Reply: Corrected.**

Page 11, line 4. Change "first" to "previous"
**Reply: Corrected.**

Page 11, line 6. Change "addations" to "addition".
**Reply: Corrected.**

Page 11, line 8. Start a new sentence at "a slight thinning area ……" and change to "An area of minor thinning ….."
**Reply: Done.**

Page 11, line 13. Repetition of "the beginning of" – delete one of them.
**Reply: Deleted.**

Page 11, line 14. Change "filed" to "field".
**Reply: Corrected.**

Page 13, line 19. Change "balances" to "balance".
**Reply: Corrected.**

Page 14, lines 8-9. Change "ice temperature of the glacier bed has not reached the melting point" to "the temperature at the glacier bed is below the melting point of ice".
**Reply: Now changed accordingly.**

Page 14, lines 12-13. Change this sentence to "In addition, internal melt caused by changes in potential energy due to glacier dynamics is negligible, as the glacier dynamics themselves are insignificant".
**Reply: Now changed accordingly.**

Page 14, line 19. Change "A calculation of" to "Equation (11) gives".
**Reply: Corrected.**

Page 14, line 20. Delete "is made".
**Reply: Deleted.**

Page 14, line 21. Correct spelling to "Storglaciären".
**Reply: Corrected.**

Page 14, line 22. Change "clod, continental" to "cold".
**Reply: Corrected.**

Page 14, lines 22-24. Repetition of a phrase – delete one of them.
**Reply: Deleted.**

Page 14, line 25. Change ", measurement and studies on internal accumulation of the is still blank" to "with no measurements of internal accumulation".
**Reply: Corrected.**

Page 14, line 28. Change "hardly" to "there is hardly".
**Reply: Corrected.**

Page 15, lines 3-4. Change "makes dynamic thinning of Urumqi Glacier No. 1" to "causes dynamic thinning of the glacier".
**Reply: Corrected.**

Page 15, line 4. Delete "make".
**Reply: Corrected.**

Page 15, line 12. Change "since" to "due to".
**Reply: Corrected.**

Page 15, line 20. Change to "Applying a reciprocal density conversion to the mass balance differences provides estimates of the ………."
**Reply: Corrected.**

Page 15, line 21. Change both "is" to "as".

**Reply: Corrected.**

Page 15, line 22. Change "from" to "the".
**Reply: Corrected.**

Page 15, line 23. Change "duo" to "due".
**Reply: Corrected.**

Page 15, line 26. Change "misalign" to "misalignment".
**Reply: Corrected.**

Page 16, line 2. Change "Despite" to "although".
**Reply: Corrected.**

Page 16, line 7. Change "relative" to "a".
**Reply: Corrected.**

Page 16, line 8. I assume that "data" should be "date".
**Reply: Thanks! Corrected.**

Page 17, line 3. Change "stakes" to "stakes can".
**Reply: Now added "can".**

Page 17, line 25. Change "has" to "have".
**Reply: Corrected.**

Page 17, line 27. Change this sentence to "Hence it should be possible to measure most glaciers using the TLS".
**Reply: Corrected.**

Page 17, line 28. Delete "of".
**Reply: Deleted.**

Page 17, line 29. Delete "number".
**Reply: Deleted.**

Page 17, line 31. Change "locate" to "are located".
**Reply: Deleted.**

Page 17, line 32. Delete "select".
**Reply: Deleted.**

Page 17, line 33. Change to "… system thus has a huge potential for glacier ……….."
**Reply: Corrected.**

Page 17, line 39. Change "annul" to "annual".
**Reply: Corrected.**

Page 17, line 40. Change sentence to "A day with little snow in the accumulation"
**Reply: Corrected.**

Page 17, line 41. Change to "… and no snow in the ablation area …"
**Reply: Corrected.**

Page 18, lines 27-28. Change "What's more" to "Further more".
**Reply: Corrected.**

[revised manuscript text omitted]
\mathrm{TLS}}}^2 = \sigma_{\Delta h\mathrm{TLS}}^2 \cdot \frac{1}{5} \cdot \frac{S_{\mathrm{cor}}}{S}, \tag{7}$$

15   where $\sigma_{\Delta h\mathrm{TLS}}$ denotes the standard deviation of TLS-derived elevation changes over stable terrain. $S_{\mathrm{cor}}$ is spatially correlated area. Given the high density ($> 1$ point m$^{-2}$) of the TLS data, we can probably assume that the number of independent items is about the number of glacier pixels (cf. Joerg et al., 2012). Here we therefore assume $S_{\mathrm{cor}} = S$. This leads to calculated values of $\sigma_{\overline{\Delta h\mathrm{TLS}}}$ range from ±0.16 to ±0.25 m (Table 2).

Uncertainties related to the density conversion for a single point ($\sigma_{\rho i}$) were calculated as
20
$$\sigma_{\rho i} = \frac{\Delta h_{\mathrm{ice}} \cdot \sigma_{\rho \mathrm{ice}} + \Delta h_{\mathrm{firn}} \cdot \sigma_{\rho \mathrm{firn}}}{\Delta h_{\mathrm{ice}} + \Delta h_{\mathrm{firn}}}, \tag{8}$$

where $\sigma_{\rho \mathrm{ice}}$ and $\sigma_{\mathrm{firn}}$ are uncertainties of ice and firn densities, which were assumed to be ±17 and ±50 kg m$^{-3}$, respectively, following Klug et al. (2018). We then extrapolated single-point values to glacier-wide uncertainties ($\sigma_{\rho}$) using the interpolation method on the ArcMAP 10.2 platform (Table 3). According to Huss et al. (2009), the uncertainties of the geodetic mass balance ($\sigma_{\mathrm{geod}}$) can be estimated using

[revised manuscript text omitted]

**Figure 7.** Spatial distributed difference derived from TLS-derived geodetic mass balance minus glaciological mass balance (a, d, g and j), black dots represent the location of well-measured ablation stakes, which are same as Fig. 5. The hypsometry (50 m altitudinal ranges) and the glaciological (dotted line) and geodetic (solid line) mass balance elevation distribution for the whole study period; both summer and annual mass balances are shown. Gray horizontal bars indicate the area-elevation distribution of Urumqi Glacier No.1. Note that the spatial resolution of glacier-wide geodetic mass balance in summer 2015 and 2016 was down-scaled to 5 m, in mass-balance years 2015-16 and 2016-17 was down-scaled to 7 m to coincide with glaciological glacier-wide mass balance.

[Figure]

**Figure 8**. Comparison between estimated and in situ measured vertical velocity for the mass balance year 2015-16 and 2016-17; the letters represent ablation stakes (Fig. 1). Note than the summer periods and stakes in the higher elevations were not selected for comparisons due to snow cover reduced the quality of in situ measured vertical velocity.

[Figure]

**Figure 9.** Changing differences between TLS-derived ($\Delta h_{TLS}$) and glaciological ($\Delta h_{glac}$) annual and summer surface elevation

10   changes at individual stakes versus normalized glacier elevation range of east branch and west branch for the four investigated

periods. Note than dash vertical lines indicate the uncertainty ranges ($\sqrt{\sigma^2_{\Delta h TLS} + (\sigma^{ice}_a)^2 + (\sigma^{firn}_a)^2}$) of the changing

differences ($\Delta h_{TLS} - \Delta h_{glac}$), and grey quadrants indicate theoretical areas for Urumqi Glacier No.1 in equilibrium.

[Figure]

**Figure 10.** Daily precipitation and mean temperature observed at Daxigou Meteorological Station during 25 April 2015 - 28 August 2017.

[Figure]

**Figure 11.** Spatial distributed slope (a) and aspect (b) of Urumqi Glacier No.1 extracted from TLS-derived DEM on 1 September 2016.

[Figure]

**Figure 12.** Spatial distribution of suitable glaciers in theory, those glaciers with an area of ≤ 1.6 km² (approximate area of Urumqi Glacier No.1) and a surface slope greater than 23.4 ° (mean slope of Urumqi Glacier No.1) have huge potential to be monitored using the TLS.

**Table 1.** Riegl VZ®-6000 TLS surveying parameters of Urumqi Glacier No.1.

| Date (dd/mm/yyyy) | Scanning range* (with overlap) (m²) | Number of points | Average point density (points m⁻²) | Vertical angle resolution (°) | horizontal angle resolution (°) | Total scan time (min) |
|---|---|---|---|---|---|---|
| 25/04/2015 | 3 204 684 | 12 740 500 | 3.98 | 0.020 | 0.020 | 46 |
| 02/09/2015 | 4 707 863 | 65 500 749 | 13.91 | 0.019/0.046 | 0.019/0.046 | 103 |
| 02/05/2016 | 3 224 285 | 26 908 210 | 8.35 | 0.020 | 0.020 | 82 |
| 01/09/2016 | 3 316 262 | 42 354 299 | 12.77 | 0.020 | 0.020 | 101 |
| 27/08/2017 | 3 161 489 | 54 835 821 | 17.34 | 0.020 | 0.020 | 88 |

*Scanning range is the total areas of four scan positions and does not include overlapped areas. The overlap percentage of the four scans on 25 April 2015 is smaller than other scan campaigns so that the average point density is relatively low.

**Table 2.** Error or StdDev ($\sigma_{MSA}$) of Multi-Station Adjustment (MSA) and the number of points ($n$) used for multi-temporal registration of two consecutive campaigns,the mean ($\mu$) and the standard error ($\sigma_{\overline{\Delta h TLS}}$) are measures of error derived by calculating elevation changes from TLS over stable terrain (off-glacier)  for 2015 summer (25 April–2 September 2015), 2015-16 (2 September 2015–1 September 2016), 2016 summer (2 May–1 September 2016) and 2016-17 (1 September 2016–27 August 2017)

| Period | Error or Stdev of MSA (m) | Number of points | Mean elevation changes over stable terrain (m) | Standard error of elevation changes over stable terrain (m) |
|---|---|---|---|---|
| 2015 (summer) | 0.28 | 11 214 842 | -0.01 | 0.25 |
| 2015-16 | 0.07 | 10 182 829 | 0.05 | 0.23 |
| 2016 (summer) | 0.20 | 10 486 985 | -0.01 | 0.22 |
| 2016-17 | 0.07 | 18 657 232 | 0.04 | 0.16 |

**Table 3.** Glacier-wide mean of density conversion (ρ) and its uncertainty ($\sigma_\rho$) (in kg m$^{-3}$) as well as TLS-derived glacier surface elevation changes ($\overline{\Delta h \text{TLS}}$) (in m). TLS-derived geodetic ($B_{\text{geod}}$) and in situ measured glaciological ($B_{\text{glac}}$) net mass balance at winter and annual scales are listed (in m w.e.).

| Period | $\rho$ | $\sigma_\rho$ | $\overline{\Delta h \text{TLS}}$ | $B_{\text{geod}}$ | $B_{\text{glac}}$ |
|---|---|---|---|---|---|
| 2015 (summer) | | | | | |
| Urumqi Glacier No.1 | 752 | 34 | -0.991 | **-0.75 ±0.19** | **-0.99 ±0.12** |
| West branch | 696 | 35 | -1.014 | -0.68 ±0.18 | -0.80 ±0.13 |
| East branch | 782 | 33 | -0.952 | **-0.82 ±0.20** | **-1.09 ±0.13** |
| 2015-16 | | | | | |
| Urumqi Glacier No.1 | 810 | 21 | -0.827 | -0.72 ±0.19 | -0.78 ±0.11 |
| West branch | 763 | 24 | -0.625 | -0.47 ±0.18 | -0.50 ±0.12 |
| East branch | 837 | 20 | -0.873 | **-0.80 ±0.19** | **-0.94 ±0.11** |
| 2016 (summer) | | | | | |
| Urumqi Glacier No.1 | 622 | 32 | -1.654 | -1.11 ±0.15 | -1.03 ±0.11 |
| West branch | 579 | 34 | -1.230 | -0.78 ±0.13 | -0.71 ±0.12 |
| East branch | 647 | 31 | -1.925 | -1.29 ±0.15 | -1.22 ±0.11 |
| 2016-17 | | | | | |
| Urumqi Glacier No.1 | 864 | 19 | -0.746 | -0.68 ±0.14 | -0. 68 ±0.11 |
| West branch | 861 | 19 | -0.844 | -0.75 ±0.14 | -0.77 ±0.11 |
| East branch | 865 | 19 | -0.729 | -0.62 ±0.14 | -0.52 ±0.11 |